# Cost-benefit analysis of coastal flood defence measures in the North Adriatic Sea

Mattia Amadio[1], Arthur H. Essenfelder[1], Stefano Bagli[2], Sepehr Marzi[1], Paolo Mazzoli[2], Jaroslav Mysiak[1], Stephen Roberts[3]

[1] *Centro Euro-Mediterraneo sui Cambiamenti Climatici, Università Ca' Foscari Venezia, Italy*

[2] *Gecosistema, Rimini, Italy*

[3] *The Australian National University, Canberra, Australia*

*Correspondence to: Arthur H. Essenfelder (arthur.essenfelder@cmcc.it)*

## Abstract

The combined effect of global sea levels rise and land subsidence phenomena poses a major threat to coastal settlements. Coastal flooding events are expected to grow in frequency and magnitude, increasing the potential economic losses and costs of adaptation. In Italy, a large share of the population and economic activities are located along the low-lying coastal plain of the North Adriatic coast, one of the most sensitive to relative sea level changes. Over the last half a century, this stretch of coast has experienced a significant rise in relative sea level, the main component of which was land subsidence; in the forthcoming decades, climate-induced sea level rise is expected to become the first driver of coastal inundation hazard. We propose an assessment of flood hazard and risk linked with extreme sea level scenarios, both under historical conditions and sea level rise projections at 2050 and 2100. We run a hydrodynamic inundation model on two pilot sites located along the North Adriatic coast of Emilia-Romagna: Rimini and Cesenatico. Here, we compare alternative extreme sea level scenarios accounting for the effect of planned and hypothetical seaside renovation projects against the historical baseline. We apply a flood damage model to estimate the potential economic damage linked to flood scenarios and we calculate the change in expected annual damage according to changes in the relative sea level. Finally, damage reduction benefits are evaluated by means of cost-benefit analysis. Results suggest an overall profitability of the investigated projects over time, with increasing benefits due to increased probability of intense flooding in the next future.

**Key-words:** coastal inundation; extreme sea level; sea level rise; cost-benefit analysis; ANUGA; Italy

**Abbreviations:** MSL (Mean Sea Level); TWL (Total Water Level); ESL (Extreme Sea Level); SLR (Sea Level Rise); VLM (Vertical Land Movements); DTM (Digital Terrain Model); EAD (Expected Annual Damage)

## 1. Introduction

Globally, more than 700 million people live in low-lying coastal areas (McGranahan et al., 2007), and about 13% of them are exposed to a 100-year return period flood event (Muis et al., 2016). On average, one million people located in coastal areas are flooded every year (Hinkel et al., 2014). Coastal flood risk shows an increasing trend in many places due to socio-economic growth (Jongman et al., 2012b; Bouwer, 2011) and land subsidence (Nicholls and Cazenave, 2010; Syvitski et al., 2009), but in the near future sea level rise (SLR) will likely be the most important driver of increased coastal inundation risk (Hallegatte et al., 2013; Hinkel et al., 2014). Evidences show that global sea level has risen at faster rates in the past century compared to the millennial trend (Church and White, 2011; Kemp et al., 2011), topping 3.6 mm per year in the last decade (2006-2015) mainly due to ocean thermal expansion and glacier melting processes (Meyssignac and Cazenave, 2012; Mitchum et al., 2010; Pörtner et al., 2019). According to the IPCC projections, it is very likely that, by the end of the 21st century, the SLR rate will exceed that observed in the period 1971-2010 for all Representative Concentration Pathway (RCP) scenarios (Pörtner et al., 2019); yet the local sea level can have a strong regional

variability, with some places experiencing significant deviations from the global mean change (Stocker et al., 2013). This is particularly worrisome in regions where small changes in the mean sea level (MSL) can drastically change the frequency of extreme sea level (ESL) events, leading up to situations where a 100-year event may occur several times per year by 2100 (Vousdoukas et al., 2018, 2017; Carbognin et al., 2009, 2010; Kirezci et al., 2020). Changes in the frequency of extreme events are likely to make existing coastal protection inadequate in many places, causing a large part of the European coasts to be exposed to flood hazard. Under these premises, coastal floods threaten to trigger devastating impacts on human settlements and activities (McInnes et al., 2003; Lowe et al., 2001; Vousdoukas et al., 2017). In this context, successful coastal risk mitigation and adaptation actions require accurate and detailed information about the characterisation of coastal flood hazard and the performance of coastal defence options. Cost-benefit analysis (CBA) is widely used to evaluate the economic desirability of a disaster risk reduction (DRR) project (Jonkman et al., 2004; Price, 2018; Mechler, 2016), helping decision-makers in evaluating the efficacy of different adaptation options (Kind, 2014; Bos and Zwaneveld, 2017).

In this study, we estimate the benefits of coastal renovation projects along the coast of Emilia-Romagna region (Italy) in terms of avoided economic losses from ESL inundation events under both current and future conditions. To do that, a range of hazard scenarios associated with ESL events are simulated over the two case study areas: i) Rimini, a touristic hotspot that is currently implementing a seafront renovation project; and ii) Cesenatico, a coastal city that could benefit from similar measures in addition to existing defence mechanisms. The scenarios are designed by combining probabilistic data from historical ESL events with the estimates of relative MSL change for those locations. Each scenario is evaluated in terms of direct economic impacts over residential areas using a flood damage model. The combination of different risk scenarios in a CBA framework allows to evaluate the economic profitability brought by the project implementation in terms of avoided losses up to the end of the century.

## 2. Area of study

Located in the central Mediterranean Sea, the Italian peninsula has more than 8,300 km of coastline, hosting around 18% of the country population, numerous towns and cities, industrial plants, commercial harbours and touristic activities, as well as cultural and natural heritage sites. Existing country-scale estimates of SLR impacts up to the end of this century helps to identify the most critically exposed coastal areas of Italy (Antonioli et al., 2017; Marsico et al., 2017; Bonaduce et al., 2016; Lambeck et al., 2011). About 40% of the country's coastal perimeter consist of a flat profile (ISPRA, 2012), potentially more vulnerable to the impacts of ESL events. The North Adriatic coastal plain is the largest and most vulnerable location to extreme coastal events due to the shape, morphology and low bathymetry of the Adriatic sea basin, which cause water level to increase relatively fast during coastal storms (Perini et al., 2017; Ciavola and Coco, 2017; Carbognin et al., 2010). Here the ESL is driven mainly by astronomical tide, ranging about one meter in the northernmost sector; and by meteorological forcing, such as low pressure, seiches and prolonged rotational wind systems, which are the main trigger of storm surge (Vousdoukas et al., 2017; Umgiesser et al., 2020). In addition to that, all the coastal profile of the Padan plain shows relatively fast subsiding rates, partially due to natural phenomena, but in large part linked to human activities (Perini et al., 2017; Carbognin et al., 2009; Meli et al., 2020). As a contributing factor to coastal flood risk, the intensification of urbanization has led to increased exposure along the Adriatic coast during the last 50 years, with many regions building over half of the available land within 300 meters from the shoreline (ISPRA, 2012). Figure **1** shows the location of the two case study areas, Cesenatico and Rimini, along with land-cover maps showing the position of coastal defences accounted in this study.

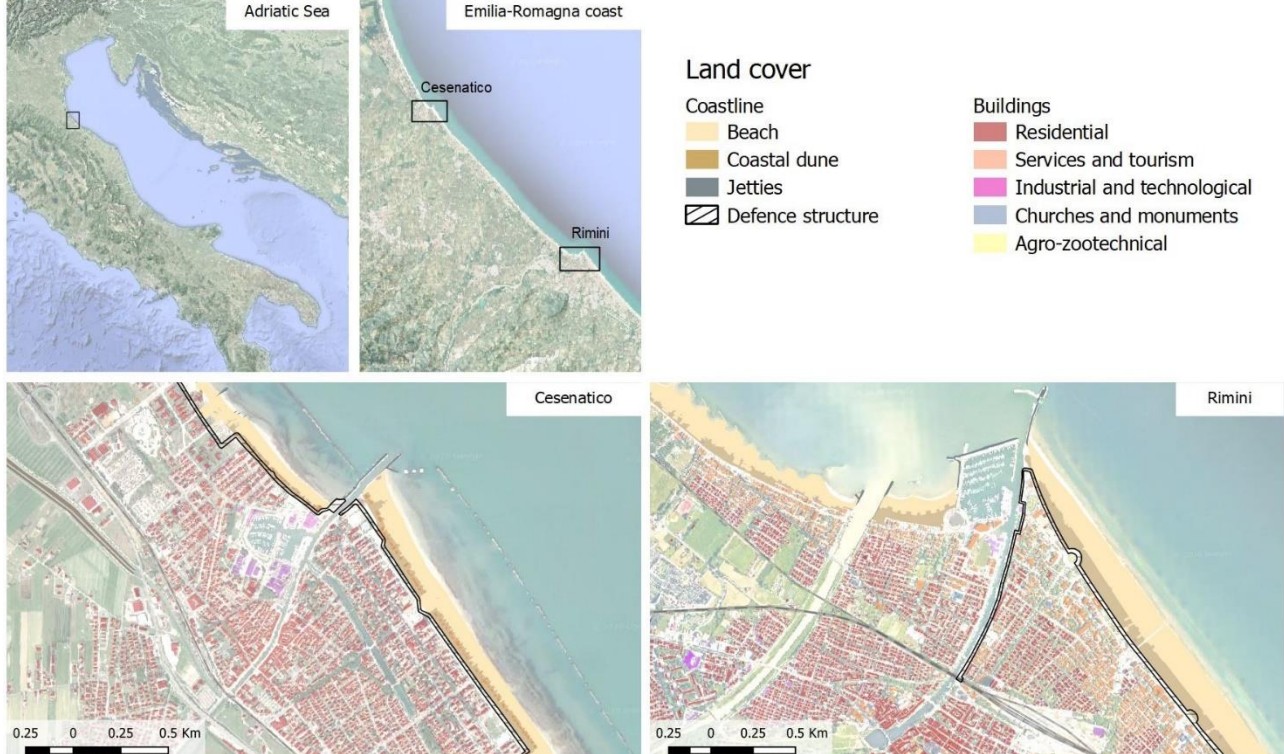

**Figure 1.** Case-study locations along the Emilia-Romagna coast: Cesenatico and Rimini. The coastal defence structure assessed in this study are shown in black. Buildings' footprint data from Regional Environmental Agency (ARPA) 2020. Basemap © Google Maps 2020.

The number of ESL events reported to cause impacts along the Emilia-Romagna coast shows a steady increase since the second half of the past century (Perini et al., 2011); this is partially explained by to the socio-economic development, which increased the extent of built-up asset potentially exposed to flood risk. The landscape along the 130 km regional coastline is almost completely flat, the only relief being old beach ridges, artificial embankments and a small number of dunes. The coastal perimeter is delineated by a wide sandy beach that is generally protected by offshore breakwaters, groins and jetties. The land elevation is often close to (or even below) the MSL, while the coastal corridor is heavily urbanised. Cesenatico has about 26,000 residents, while Rimini has 150,000. The towns have a strong touristic vocation, hosting large beach resort and bathing facilities along the beach and hundreds of hotels and rental housing located just behind the seaside. Both places have been affected by coastal storms resulting in flooding of buildings and activities, beach erosion and regression of the coastline. The most recent inundation events were observed in March 2010, November 2012 and February 2015. The 2015 event was one of the most severe ever recorded, with ESL values corresponding to a probability (return period) of once in 100 years. It caused severe damages along the whole regional coast and, in some locations, required the evacuation of people from their houses; many buildings and roads were covered by sand brought by the flood wave; touristic infrastructures near the shore were seriously damaged, and some port channels overflowed the surrounding areas. The economic impact of the event was estimated topping 7.5 M Eur (Perini et al., 2015).

## 3. Methodology

### 3.1 Components of the analysis

Coastal inundation is caused by an increase of total water level (TWL), most often associated to extreme sea level (ESL) events, which are often generated by a combination of high astronomical tide and meteorological

drivers such as storm surge and wind waves (Figure **2**). Probabilistic flood risk assessments generally consider
ESL as the result of the combined effects of storm surge and tides (Muis et al., 2015; Vousdoukas et al., 2017).
More recent studies also account for the effects of waves by either adding wave setup to the ESL or by
simulating the dynamics of breaking waves on the coast (Kirezci et al., 2020; Melet et al., 2020; Li et al., 2020;
Wang et al., 2021; Muis et al., 2020; Lionello et al., 2021; McInnes et al., 2009; Idier et al., 2019). In our study,
we consider TWL near shore as the result of the combination of high tide, storm surge, and wave action, the
latter combining wave setup (defined as the increase of mean sea level at the shore that is caused by the loss
of wave momentum in the surf zone) with wave periodicity of incoming breaking waves (which defines wave
swash, i.e. the amplitude of the time varying elevation due to breaking waves along the shore).

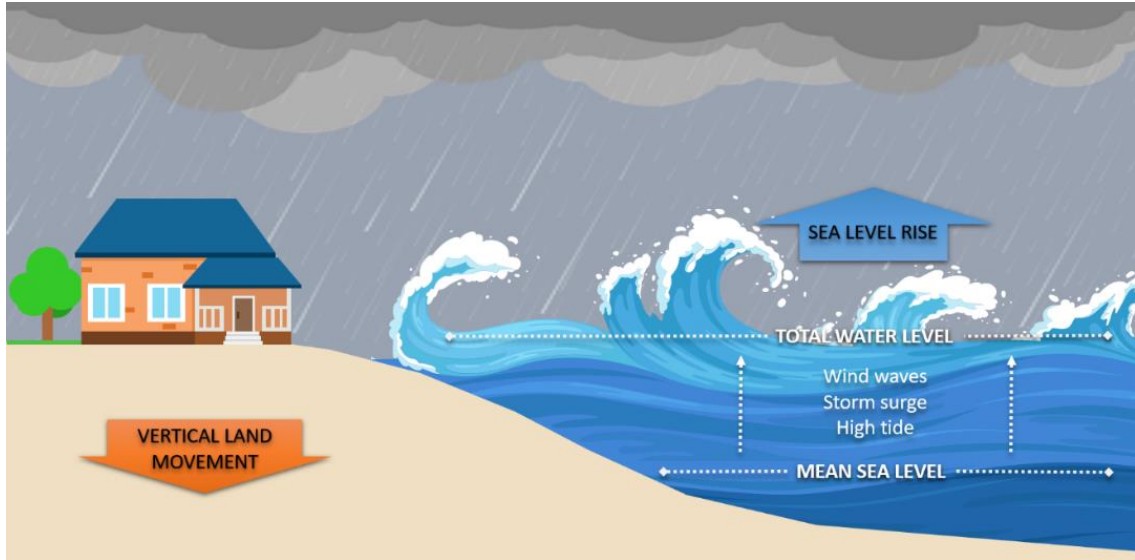

**Figure 2.** Components of the analysis for extreme sea level events: total water level is the sum of maximum
tide, storm surge and wind waves over mean sea level. Vertical land movement and sea level rise affects the
mean sea level on the long run.

The identification of areas threatened by coastal flooding from ESL events is often done by means of flood
maps, which are generated through hydrostatic or hydrodynamic modelling approaches. These approaches
differ substantially in their complexity and their ability to represent environmental processes. The hydrostatic
inundation approach (sometimes referred as "bathtub") is methodologically simple and computationally
quick, as it does not consider dynamic processes such as flow mass conservation and the effect of land cover
on the spread of floodwater, assuming flooded areas as those with an elevation below than a forcing water
level (Hinkel et al., 2010, 2014; Jongman et al., 2012b; Ramirez et al., 2016; Vousdoukas et al., 2016; Muis et al.,
2016). These assumptions and simplifications often result in substantial misestimation of flood extents
compared to the hydrodynamic flood modelling and observations (Bates et al., 2005; Vousdoukas et al., 2016;
Breilh et al., 2013; Ramirez et al., 2016; Seenath et al., 2016; Kumbier et al., 2019; Anderson et al., 2018). To
overcome these limitations, hydrodynamic flood modelling approaches capable of accounting for the effects
of wind, waves, tide, current, and river run-off can be used (Barnard et al., 2019). The most advanced models
can simulate atmospheric–ocean–land interactions from the deep ocean to the coast with a satisfactory
predictive skill (Bates et al., 2005; Seenath et al., 2016; Vousdoukas et al., 2016; Lewis et al., 2013), at the costs
of a more complex model setup, extensive data requirements and significantly longer computational times
(Teng et al., 2017). As intermediate solution, simplified hydrodynamic flood models that focus on nearshore
processes are capable of reducing the computational cost while taking into consideration water mass
conservation (Breilh et al., 2013), aspects of flooding hydrodynamics (Dottori et al., 2018) and the presence of
obstacles (Perini et al., 2016). They proved to be reliable for coastal flooding applications, such as the
simulation of coastal flooding due to storm-tide events (Ramirez et al., 2016; Bates et al., 2005; Skinner et al.,
2015; Smith et al., 2012).
In this study, estimates of ESL components (storm surge, tides and waves) are obtained for the North Adriatic
up to year 2100 by combining reference hazard scenarios derived from the analysis of historical records (Perini
et al., 2011, 2016, 2017; Armaroli et al., 2012; Armaroli and Duo, 2018) with regionalised projections of SLR
(Vousdoukas et al., 2017) and local vertical land movements (VLM) rates (Perini et al., 2017; Carbognin et al.,
2009). On this basis, four hypothetical ESL scenarios are designed, ranging from low intensity-high frequency
to high intensity-low frequency, under both current and future (2050 and 2100) conditions. The hydrodynamic
model ANUGA (Roberts et al., 2015) is applied to simulate the inundation of land areas during ESL accounting
for individual components. Land morphology and exposure of coastal settlements are described by high-
resolution DTM (LiDAR) and bathymetry, in combination with land use and buildings footprints. The effect
of hazard mitigation structures (both designed and under construction) are explicitly accounted by the model
in the "defended" scenario, in contrast to the baseline scenario, where only existing defence structures (groins,
jetties, breakwaters and sand dunes) are considered.

## 3.2 Vertical Land Movement

Vertical land movements result from a combination of slow geological processes such as tectonic activity and
glacial isostatic adjustment (Peltier et al., 2015; Peltier, 2004), and medium-term phenomena, such as sediment
loading and soil compaction (Carminati and Martinelli, 2002; Lambeck and Purcell, 2005). The latter can
greatly oversize geological processes at local scale (Wöppelmann and Marcos, 2012); in particular, faster
subsidence occurs in presence of intense anthropogenic activities such as water withdrawal and natural gas
extraction (Teatini et al., 2006; Polcari et al., 2018). Most of the peninsula shows a slow subsiding trend,
although with some local variability. An estimate of VLM rates due to tectonic activity has been derived from
studies conducted in Italy (Solari et al., 2018; Antonioli et al., 2017; Marsico et al., 2017; Lambeck et al., 2011).
The North Adriatic coastal plain shows the most intense long-term geological subsidence rates (about 1 mm
per year), increasing North to South. Yet in the last decades these rates were often greatly exceeded by ground
compaction rates observed by multi-temporal SAR Interferometry (Gambolati et al., 1998; Antonioli et al.,
2017; Polcari et al., 2018; Solari et al., 2018). Observed subsidence is about one order of magnitude faster where
the aquifer system has been extensively exploited for agricultural, industrial and civil use since the post-war
industrial boom. From the 1970s, however, with the halt of groundwater withdrawals, anthropogenic drivers
of subsidence has been strongly reduced or stopped (Carbognin et al., 2009). Nonetheless, subsidence still
continues at much faster rates than expected from natural phenomena (Teatini et al., 2005). Geodetic surveys
carried out from 1953 to 2003 along the Ravenna coast provide evidence of a cumulative land subsidence
exceeding 1 m at some sites due to gas extraction activities. Average subsidence rates observed for 2006-2011
along the Emilia-Romagna coast are around 5 mm/yr, exceeding 10 mm/yr in the back shore of the Cesenatico
and Rimini areas and topping 20-50 mm/yr in Ravenna (Perini et al., 2017; Carbognin et al., 2009). Based on
these current rates, we assume an average fixed annual VLM of 5 mm in both Cesenatico and Rimini up to the
end of the century. This remarkable difference between natural VLM rates and observations would produce a
dramatic effect on the estimated SLR scenarios: at present rates, Rimini would see an increase of MSL by 0.15
m in 2050 and more than 0.4 m in 2100 independently from eustatic SLR. Since these rates are connected with
human activity, it is not possible to foresee how they will change in the longer run.

## 3.3 Sea Level Rise

The long availability of tide gauge data along the North Adriatic coast allows to assess the changes in MSL
during the last century, estimated to be +1.3 mm/year (Scarascia and Lionello, 2013). This is consistent with
published values for the Mediterranean Sea (Tsimplis et al., 2008; Tsimplis and Rixen, 2002) and the Adriatic
Sea (Tsimplis et al., 2012; Carbognin et al., 2009). The projections of future MSL account for sea thermal
expansions from four global circulation models, estimated contributions from ice-sheets and glaciers (Hinkel
et al., 2014) and long-term subsidence projections (Peltier, 2004). The ensemble mean is chosen to represent
each RCP for different time slices. The increase in the central Mediterranean basin is projected to be
approximately 0.2 m by 2050 and between 0.5 and 0.7 m by 2100, compared to historical mean (1970-2004)
(Vousdoukas et al., 2017). As agreed with local stakeholders (*Comune di Rimini*), our analysis considers the
intermediate emission scenario RCP 4.5, projecting an increase in MSL of 0.53 m at 2100. It must be noted that
these projections, although downscaled for the Adriatic basin, do not account for the peculiar continental
characteristics of the shallow northern Adriatic sector, where the hydrodynamics and oceanographic
parameters partially depend on the freshwater inflow (Zanchettin et al., 2007).

### 3.4    Tides and meteorological forcing

Storm surge and wind waves represent the largest contribution to TWL during an ESL event. An estimation
of  these components is obtained for the two coastal sites from the analysis of tide gauge and buoy records,
and from the description of historical extreme events presented in local studies (Armaroli and Duo, 2018;
Perini et al., 2012; Masina et al., 2015; Perini et al., 2011, 2017). This area is microtidal: the mean neap tidal
range is 30–40 cm, and the mean spring tidal range is 80–90 cm. Most storm surge events have a duration of
less than 24 h and a maximum significant wave height of about 2.5 m. During extreme cyclonic events, the
sequence of SE wind (*Sirocco*) piling the water North and E-NE wind (*Bora*) pushing waves towards the coast
can generate severe inundation events, with significant wave height ranging 3.3 – 4.7 m and exceptionally
exceeding 5.5 m (Armaroli et al., 2012). Fifty significant events have been recorded from 1946 to 2010 on the
ER coast, with half of them causing severe impacts along the whole coast and 10 of them being associated with
important flooding events (Perini et al., 2017). The most severe events are found when strong winds blow
during exceptional tide peaks, most often happening in late autumn and winter. The event of November 1966
represents the highest ESL on records, causing significant impacts along the regional coast: the recorded water
level was 1.20 m above MSL, and wave heights offshore were estimated around 6–7 m (Garnier et al., 2018;
Perini et al., 2011). The whole coastline suffered from erosion and inundation, especially in the province of
Rimini. Atmospheric forcing shown significant variability for the period 1960 onwards (Tsimplis et al., 2012),
but there is no strong evidence supporting a significant change in marine storminess frequency or severity for
the near future (Lionello, 2012; Zanchettin et al., 2020; Lionello et al., 2020, 2017). Thus, in our model we assume
meteorological forcing to remain the same up to 2100.

### 3.5    Terrain morphology and coastal defence structures

Reliable bathymetry and topography are required in order to run the hydrodynamic modelling at the local
scale. Bathymetric data for the Mediterranean Sea were obtained from the European Marine Observation and
Data Network (EMODnet) at 100 m resolution. The description of terrain morphology comes from the official
high-resolution LiDAR DTM (MATTM, 2019). First, we combined the coastal dataset (2 m resolution and
vertical accuracy of ± 0.2 m), and the inland dataset (1 m resolution and vertical accuracy ± 0.1 m) into one
seamless layer. Then, the DTM is supplemented with geometries of existing coastal protection elements such
as jetties, groins and breakwaters obtained from the digital Regional Technical Map. In Rimini, the *Parco del*
*Mare* (Figure **3**) is an urban renovation project which aims to improve the seafront promenade: the existing
road and parking lots are converted into an urban green infrastructure consisting of a concrete barrier covered
by vegetated sandy dunes with walking paths. This project also acts as a coastal defence system during
extreme sea level events. The barrier rises 2.8 meters along the southern section of the town, south of the

marina; no barrier is planned on the northern coastal perimeter. The *Parco del Mare* project is expected to be completed by 2021 and has been taken in account in the evaluation of the "defended" scenarios by adding the barrier elevation to the DTM.

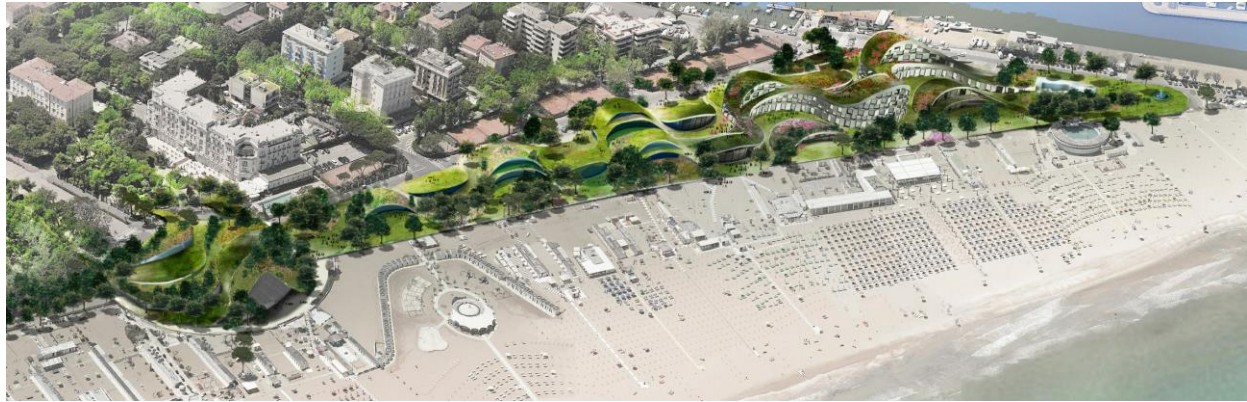

**Figure 3.** Prototype design of Parco del Mare project in Rimini. Adapted from JDS Architects.

In Cesenatico, the existing defence structures include a moving barrier system (*Porte Vinciane*) located on the port channel, coupled with a dewatering pump which discharge the meteoric waters in the sea. The barriers close automatically if the TWL surpasses 1 meter over the mean sea level, preventing floods in the historical centre up to 2.2 meters of TWL. Additional defence structures include the winter dunes, which consist of a 2.2 meter-tall intermittent, non-reinforced sand barrier. In the defended scenario, we envisage a coastal defence structure similar to Rimini's *Parco del Mare* project, spanning both North and South of the port channel with a total length of 7.8 km. The DTM was manually edited based on additional reference data (i.e. on-site observations or aerial photography) in order to remove artefacts and to produce a more realistic representation of the land morphology. Bridges and tunnels are the most critical elements that required DTM correction in order to avoid misrepresentations of the water flow routing.

### 3.6    Scenario design

In order to design probabilistic nearshore scenarios associated to ESL of different intensities to use as boundary conditions in the hydrodynamic model, we rely on existing analysis of ESL events occurring on the regional coast (Perini et al., 2011, 2016, 2017), which have been adopted by the Regional Environmental Agency to define the official coastal flood hazard zones and related protection standards (ARPA Emilia-Romagna, 2019). The probability of occurrence of these ESL scenarios is expressed in terms of return period (*RP*), which is the estimated average time interval (in years) between events of similar intensity. Four scenarios of increasing intensity are designed, namely *RP 1, 10, 100* and *250* years. For each of these hypothetical scenarios, the TWL nearshore is calculated as the sum of extreme values for storm surge level (*SS*), max tide (*Tmax*) and wave action contribution (*Wc*) at each time-step (see Table **1**). In particular, given the limitation of the considered 2D hydrodynamic model in not resolving vertical convection and waves breaking (i.e. swash), we include wave action contribution to ESL by accounting for wave setup (*Ws*) with a periodicity equal to the incoming breaking waves (*Wp*), thus partially representing wave motion (Armaroli et al., 2012, 2009). We develop a set of trigonometric equations based on harmonic analysis concepts to characterise the amplitude and period of tidal, storm surge, and wave levels as the harmonics constituents that describe the theoretical temporal evolution of the nearshore TWL during an ESL event (see Figure **4**). Harmonics constituents are the elements in a mathematical expression of a series of periodic terms and have been used in harmonic analysis for sea level prediction (Boon, 2011; Familkhalili et al., 2020; Fuhrmann et al., 2019; Annunziato and Probst, 2016). The set of equations used in the study are specified in Annex 1, together with sample applications and validation metrics to observed ESL events along the coast of ER. Additional variables to characterize the event dynamics

are the storm surge duration (*Time*, in hours) and the wave period (*Wp*, in seconds), both obtained from regional studies of ESL events (Armaroli et al., 2012; Armaroli and Duo, 2018). Projections of TWL at 2050 and 2100 are calculated for the same set of RP scenarios by adding SLR and VLM contributions to the MSL, thus shifting the TWL curve up by 33 cm in 2050 and by 97 cm in 2100.

**Table 1.** components of nearshore TWL for four ESL scenarios (RPs) designed according to analysis of historical ESL events and projected MSL change (2050 and 2100), accounting for both SLR (RCP 4.5) and average VLM rate.

| | Extreme event features | | | | | Historical | 2050 | | | 2100 | | |
|---|---|---|---|---|---|---|---|---|---|---|---|---|
| *RP* | *SS* | *Tmax* | *Ws* | *Time* | *Wp* | *TWL* | *SLR* | *VLM* | *TWL* | *SLR* | *VLM* | *TWL* |
| (years) | (m) | (m) | (m) | (h) | (s) | (m) | (m) | (m) | (m) | (m) | (m) | (m) |
| 1 | 0.60 | 0.40 | 0.22 | 32 | 7.7 | 1.22 | 0.14 | 0.19 | 1.55 | 0.53 | 0.44 | 2.19 |
| 10 | 0.79 | 0.40 | 0.30 | 42 | 8.9 | 1.49 | 0.14 | 0.19 | 1.82 | 0.53 | 0.44 | 2.46 |
| 100 | 1.02 | 0.40 | 0.39 | 55 | 9.9 | 1.81 | 0.14 | 0.19 | 2.14 | 0.53 | 0.44 | 2.78 |
| 250 | 1.40 | 0.45 | 0.65 | 75 | 11 | 2.50 | 0.14 | 0.19 | 2.83 | 0.53 | 0.44 | 3.47 |

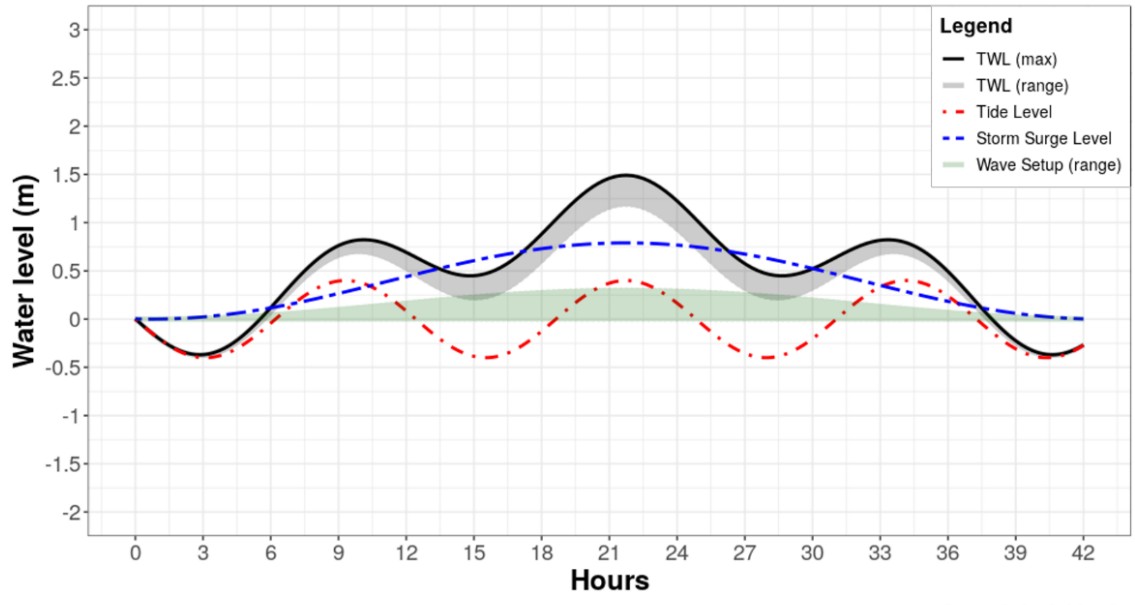

Source: Authors' elaboration

**Figure 4.** Design of dynamic ESL scenario corresponding to RP 10 years under historical MSL conditions. The maximum TWL is shown as the black continuous line, while TWL range at any given time is shown as the shaded grey area. The components of the nearshore TWL are the tide level (red dashed line), the storm surge level (blue dashed line) and the wave action contribution (green shaded area). Wave action contribution is represented as a shaded area due to its high frequency (period of 8.9s). In this scenario (RP 10) the maximum storm surge level is 0.79 m, the maximum high tide is 0.40 m, and the wave action contribution ranges from 0.00 m to 0.30 m, with a wave period of 8.9 seconds. At the peak of the event, these conditions produce a maximum TWL of 1.49 m.

Figure **5** shows how the nearshore TWL results at any given time from the combination of storm surge, tide level and wave action contribution in the scenario RP 10 years (additional figures for all RP scenarios can be found in Annex 1). The individual contribution of *SS* and *Tmax* levels are represented by coloured dashed lines in the figure. The *Wc* component is shown as a green shaded area due to its high frequency (defined by the wave period, *Wp*, in seconds), thus representing the range of values assumed in any given time. The intensity of waves contribution to ESL is assumed to grow proportionally to the increase of the *SS* component. The shaded grey area represents the range of TWL as sum of these components, while the black continuous

line represents the maximum TWL at any given time. Our approach is precautionary as it provides worst-case
TWL values: *SS* peak is set to coincide with *Tmax* and *Wc* at the mid of the event, thus resulting in the
maximum TWL possible under each scenario.

### 3.7 Inundation modelling

The nearshore ESL scenarios specified in Table 1 and exemplified in Figure 4 (and Annex 1) are used as forcing
boundary condition in ANUGA, a 2D hydrodynamic model suitable for the simulation of flooding resulting
from riverine peak flows and storm surges (Roberts, 2020). The fluid dynamics simulation is based on a finite-
volume method for solving the shallow water wave equations, thus being based on continuity and simplified
momentum equation. Being a 2D hydrodynamic model, ANUGA does not resolve vertical convection, waves
breaking or 3D turbulence (i.e. vorticity), thus it cannot account for the swash component of wave runup.
Wave direction is set to be oriented perpendicular to the coast. For each scenario, ANUGA computes the TWL
on the coast, the resulting water depth of inundation, and the horizontal momentum on an unstructured
triangular grid (mesh) representing the two case study areas. The size of the triangles is variable within the
mesh, thus allowing or a better representation in regions of particular interest, such as along the coastline, in
urban areas, and inside the canals. Six different regions are used in each case study to define different
triangular mesh resolution, varying from higher resolution areas of 16 m² for canals and coastal defence
structures, to lower resolution of 900 m² for sea areas. The output of the simulation consists of maps
representing flood extent, water depth and momentum at every time step (~1 second), projected on the high-
resolution DTM grid (1 meter). Figure **6** presents the two case study areas and the respective resolutions for
each region. The resulting irregular mesh counts with about 637 thousand triangles for the Cesenatico domain,
and about 1,2 million triangles for the Rimini domain. The model includes an operator module that simulates
the removal of sand associated with over-topping of a sand dune by sea waves. The operator simulates the
erosion, collapse, fluidisation and removal of sand from the dune system (Kain et al., 2020). This option is
enabled only in the undefended scenario for Cesenatico, where non-reinforced sand dunes are prone to
erosion.

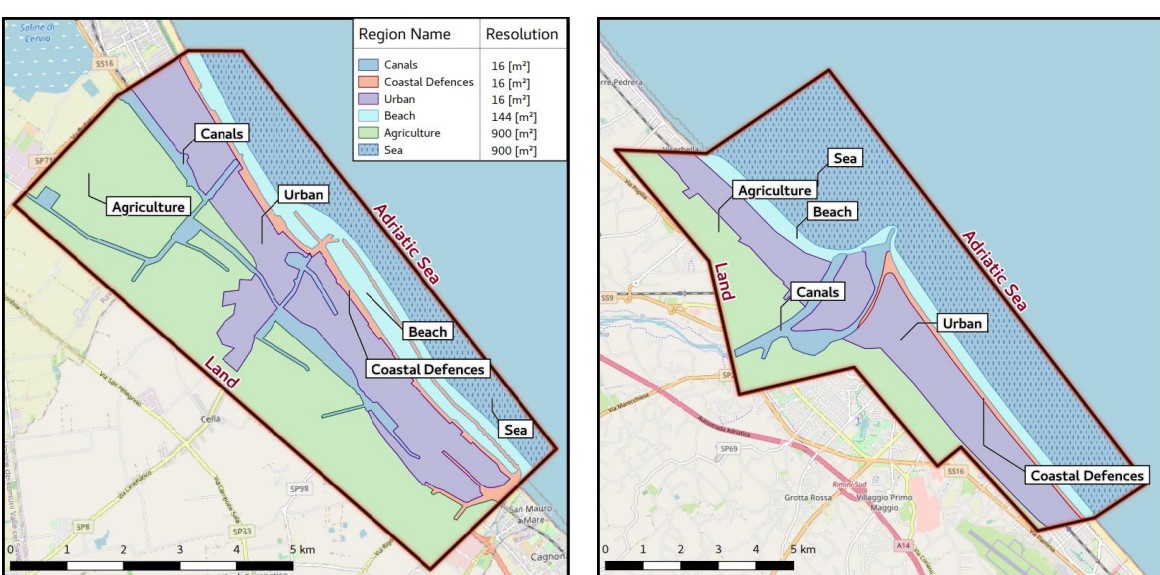

**Figure 5.** The definition of simulation domain for the cities of Cesenatico (on the left) and Rimini (on the right).
The legend shows the mesh resolution specific to each region simulated by the model.

## 3.8 Risk modelling and Expected Annual Damage

Direct damage to physical asset is estimated using a customary flood risk assessment approach originally developed for fluvial inundation, which is adapted to coastal flooding assuming that the dynamic of impact from long-setting floods depends on the same factors, namely: 1) hazard magnitude, and 2) type, size and value of exposed asset. Indirect economic losses due to secondary effects of damage (e.g. business interruption) are excluded from the computation. Hazard magnitude can be defined by a range of variables, but the most important predictors of damage are water depth and the extension of the flood event (Jongman et al., 2012a; Huizinga et al., 2017). The characterization of exposed asset is built from a variety of sources, starting from land use and buildings footprints obtained from the Regional Environmental Agencies geodatabases and the Open Street Map database (Open Street Map data for Nord-Est Italy, 2019). Additional indicators about buildings characteristics are obtained from the database of the 2011 Italian Census (15∘ censimento della populazione e delle abitazioni, 2019), while mean construction and restoration costs per building types are obtained from cadastral estimates (CRESME, 2019). The asset representation is static, thus not accounting for changes in land use nor population density, while allowing for the direct comparison of hazard mitigation options' results. A depth-damage function validated on empirical records (Amadio et al., 2019) is applied in order to translate each hazard scenario into an estimate of economic risk, measured as a share of total exposed value. The damage function applies only to residential and mixed-residential buildings, the area of which represents about 93% of total exposed footprints; other types (such as harbour infrastructures, industrial, commercial, historical monuments and natural sites) are excluded from risk computation. Abandoned or under-construction buildings are also excluded from the analysis. To avoid overcounting of marginally-affected buildings, we set two threshold conditions for damage calculation: flood extent must be greater than or equal to 10 m², and maximum water depth must be greater than or equal to 10 cm. The damage/probability scenarios are combined together as Expected Annual Damage (EAD). EAD is the damage that would occur in any given year if damages from all flood probabilities were spread out evenly over time; mathematically, EAD is the integration of the flood risk density curve over all probabilities (Olsen et al., 2015), as in equation **1**.

$$EAD = \int_0^1 D(p)\, dp \tag{1}$$

The integration of the curve can be solved either analytically or numerically, depending on the complexity of the damage function $D(p)$. Several different methods for numerical integration exist; we use an approach where EAD is the sum of the product of the fractions of exceedance probabilities by their corresponding damages (Figure **7**). We calculate $D(p)$, which is the damage that occurs at the event with probability $p$, by using the depth-damage function for each hazard scenario. The exceedance probability of each event ($p$) is calculated based on exponential function as shown in equation **2**.

$$p = 1 - e^{\left(\frac{-1}{RP}\right)} \tag{2}$$

Events with a high probability of occurrence and low intensity (below RP 1 year) are not simulated, as they are assumed to not cause significant damage. This is consistent with the historical observations for the case study area, although this assumption could change with increasing MSL.

**Figure 6.** Schematic representation of the numerical integration of the damage function $D(p)$ with respect to the exponential probability of the hazard events. Damage (Y axis) represents the ratio of damage to the total exposed value estimated up to the most extreme scenario (RP 250 years). Events with a probability of occurrence higher than once in a year are expected to not cause damage (grey area).

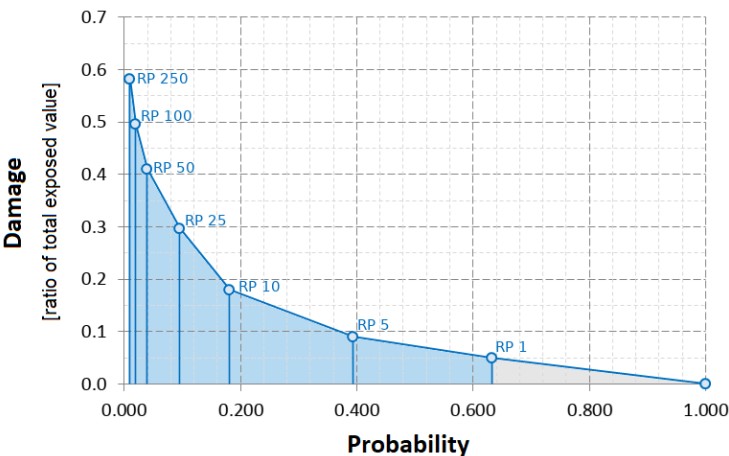

### 3.9 Cost-Benefit Analysis

A CBA should include a complete assessment of the impacts brought by the implementation of the hazard mitigation option, i.e. direct and indirect, tangible and intangible impacts (Bos and Zwaneveld, 2017). The project we are considering, however, has not been primarily designed for DRR purpose: instead, it is meant as an urban renovation project which aims to consolidate the touristic vocation of the area, to improve the quality of life and the urban environment (Comune di Rimini, 2018). This implies some large indirect effects on the whole area, most of which are not strictly related to disaster risk management and, overall, very difficult to estimate ex-ante. Our evaluation focuses only on the benefits that are measurable in terms of direct flood losses reduction. Regarding the implementation costs, the CBA accounts for the initial investment required for setting up the adaptation measure, and operational costs through time. According to the *Parco del Mare* project funding documentation (Comune di Rimini, 2019b, a, 2020, 2021a, b), the total cost of the project (to be completed during 2021) is 33.3 M Eur, corresponding to 5.55 M Eur per Km of length. No information is available about maintenance costs of the opera, but given the nature of the project (static defense with low structural fragility), we assume they will be rather small compared to the initial investment. Ordinary annual maintenance costs are accounted as 0.1% of the total cost of the project. The same costs are assumed for the hypothetical barrier in Cesenatico, resulting in an initial investment cost of 43.3 M. Costs and benefit occurring in the future periods need to be discounted, as people put higher value on the present (Rose et al., 2007). This is done by adjusting future costs and benefits using an annual discount rate ($r$). We chose a variable rate of $r = 3.5$ for the first 50 years and $r = 3$ from 2050 onward (Lowe, 2008). A sensitivity analysis of discount rate is included in Annex 2. The three main decision criteria used in CBA for project evaluation are the Net Present Value (NPV), the Benefit/Cost Ratio (BCR) and the payback period. The NPV is the sum of Expected Annual Benefits ($B$) up to the end of the time horizon, discounted, minus the total costs for the implementation of the defense measure, which takes into account initial investment plus discounted annual maintenance costs ($C$). In other words, the NPV of a project equals the present value of the net benefits ($NB_i = B_i - C_i$) over a period of time (Boardman et al., 2018), as in equation **3**:

$$NPV = PV(B) - PV(C) = \sum_{t=0}^{n} \frac{NB_r}{(1+r)^t} \tag{3}$$

Positive NPV means that the project is economically profitable. The BCR is instead the ratio between the benefits and the costs; a BCR larger than 1 means that the benefits of the project exceed the costs on the long term and the project is considered profitable. The payback period is the number of years required for the discounted benefits to equal the total costs.

## 4. Results

### 4.1 Inundation scenarios

Once the setup is completed, the hydrodynamic model performs relatively fast: each simulation is carried at half speed compared to real time, requiring about 24 hours to simulate a 12 h event. Parallel simulations for the same area can run on a multicore processor, improving the efficiency of the process. The output of the hydrodynamic model consists of a set of inundation simulations that include several hazard intensity variables in relation to flood extent: water depth, flow velocity, and duration of submersion. ESL scenarios are then summarized into static maps, each one representing the maximum value reached by hazard intensity variables during the simulated event at about 1 meter resolution. The flood extents corresponding to each RP scenario are shown for Rimini (Figure 8) and Cesenatico (Figure 9).

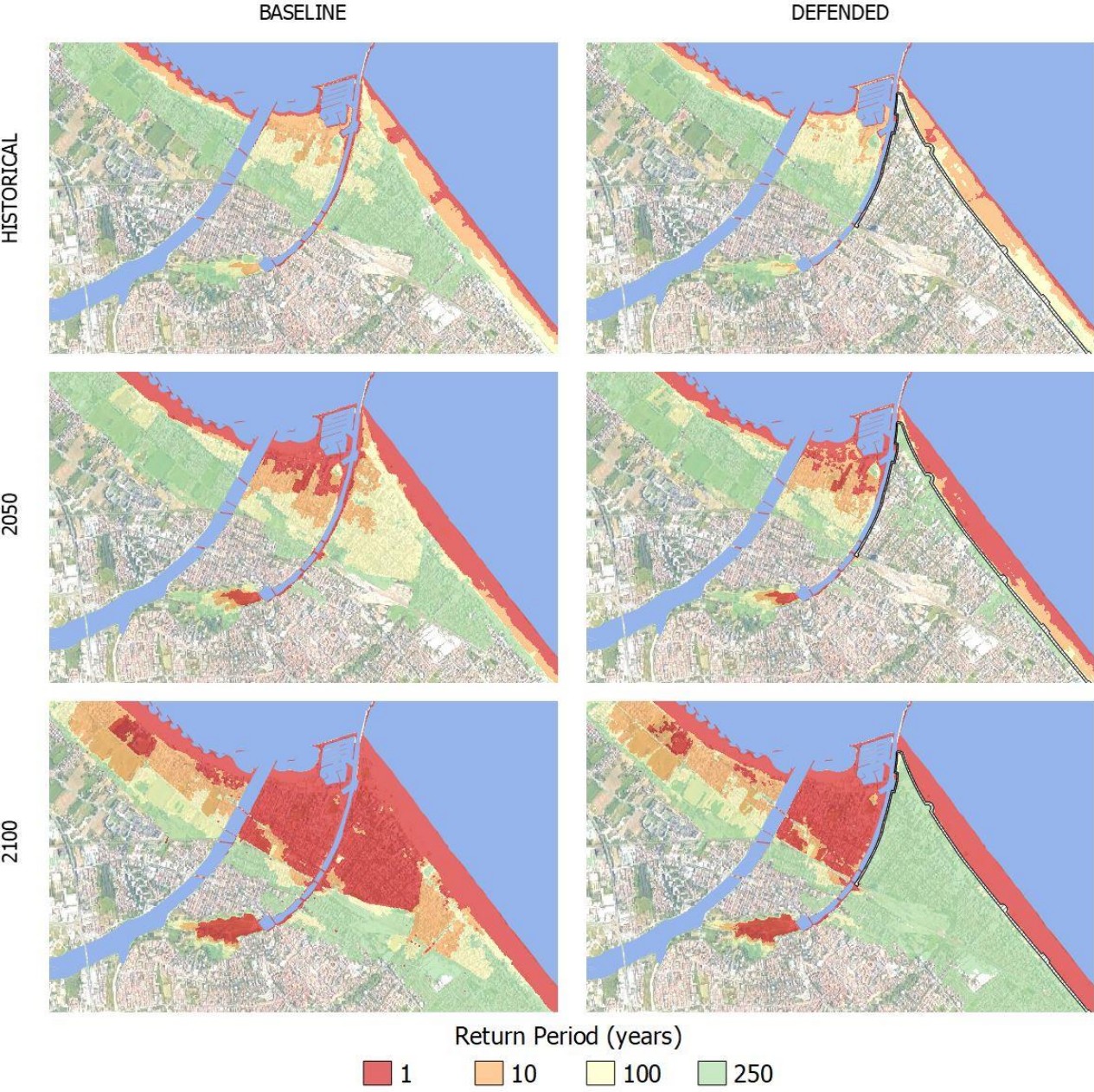

**Figure 7.** Rimini, extent of land affected by flood according to frequency of occurrence of ESL event up to 2100 for the baseline [left] and the defended scenario [right]. Basemap © Google Maps 2020.

In Rimini, the *Parco del Mare* barrier produces benefits in terms of avoided flooding in the south-eastern part of the town (high-density area) for ESL events with a return period of 100 years or less. The north-western part and the marina are outside of the defended area; these areas are therefore subject to a similar amount of flooding across scenarios. In all the simulations, the buildings located behind the marina are the firsts to be flooded. In fact, the new and the old port channels located on both sides of the marina represent a hazard hotspot: as shown in the maps, the failure of the eastern channel, which has a relatively low elevation, is likely to cause the water to flood the eastern part of the town, even during inundation events that would not surpass the beach. In the defended scenarios, where both the coastal and the canal barriers are enabled, the flood extent in the south-eastern urban area becomes almost zero for ESL events with a probability of once in 100 years, even when accounting for SLR up to 2100. Under the most exceptional ESL conditions (RP 250 in 2100), the barrier is overtopped, generating a flood extent similar to the baseline scenario for the same occurrence probability.

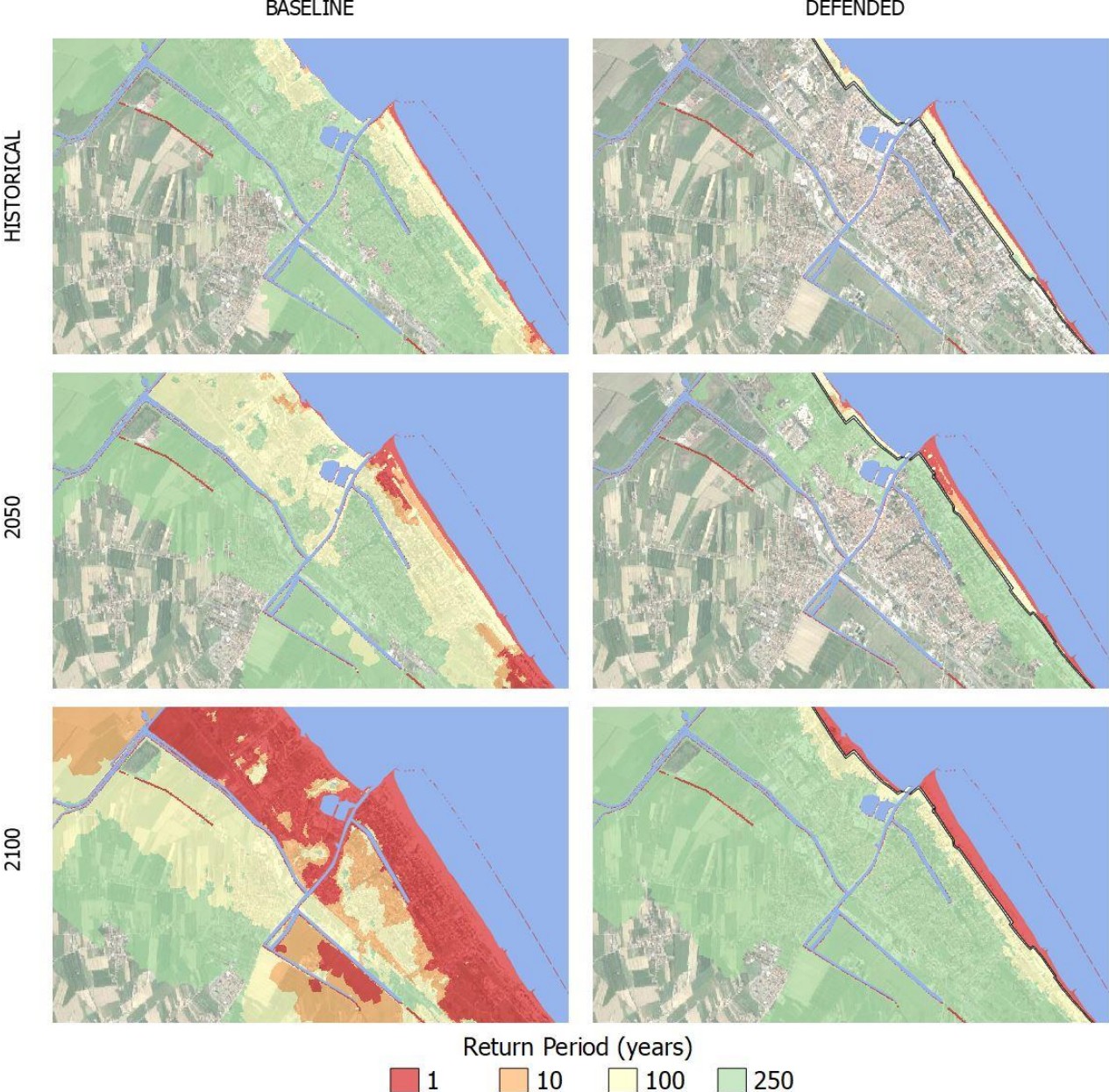

**Figure 8.** Cesenatico, extent of land affected by flood according to frequency of occurrence of ESL event up to 2100 for the baseline [left] and the defended scenario [right]. Basemap © Google Maps 2020.

In Cesenatico, a barrier designed similarly to *Parco del Mare* could provide significant reduction of flood extents under most hazard scenarios. Its effectiveness would be greater than in Rimini thanks to the complementary movable barrier system in use, which seals the port channel allowing to wall off the whole coastal perimeter, reducing the chance of water ingression in the urban area. In contrast, the erodible winter dune in the baseline defense scenario can only hold the heavy sea for shorter, less intense ESL events (RP 1 – 10 years), and becomes ineffective with more exceptional, long-lasting events; from 2050 on, the winter dune could be surmounted and dismantled by sea waves even during non-exceptional events (RP 1 year).

### 4.2 Expected Annual Damage

The Expected Annual Damage is calculated as a function of maximum exposed value and water depth. In Rimini, the EAD grows from around 650 thousand Eur under historical conditions to 2.8 million Eur in 2050 and more than 32.3 million Eur in 2100. Under less severe ESL scenarios (RP below 100 years), the risk remains mostly confined around the marina, which is located outside the defended area, producing an expected damage below 10 thousand Eur. Under more extreme ESL scenarios, the benefits of the *Parco del Mare* project protecting the southern part of Rimini become more evident, avoiding about 65% of the expected damages in the defended scenarios compared to the undefended ones. The damage avoided in the defended scenarios grow almost linearly with the increase of the baseline EAD under future projections of sea level rise: under the defended scenario, the EAD is reduced on average by 45% in comparison with the undefended scenario (Figure **10**, left). The project produces benefit up to scenario RP 250 years in 2100, where a projected TWL of 3.5 meters would cause the overtopping of the barrier, reducing the benefits to almost zero (Figure **9**, right).

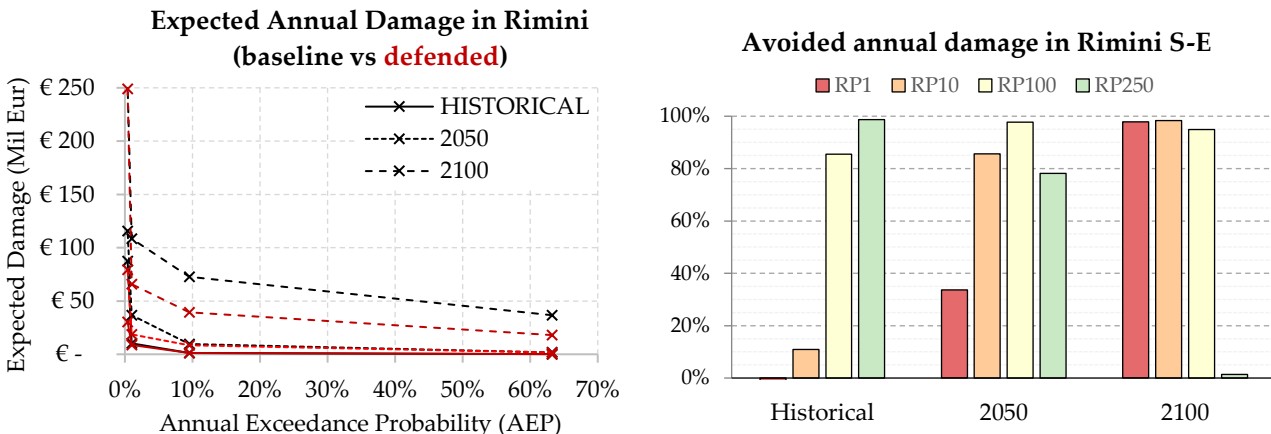

**Figure 9.** Rimini: Expected Annual Damage (EAD) according to undefended scenario up to 2100, all town considered [left]; EAD reduction in the south-eastern part of the town thanks to hazard mitigation offered by the coastal barrier [right].

In Cesenatico, the average EAD for the undefended scenario grows from around 270 thousand Eur under historical conditions, to 1.7 million Eur in 2050 and almost 26 million Eur in 2100. In our simulations, the designed defence structure (a static barrier with height of 2.8 m along 7.8 km of coast) is able to avoid most of the damage inflicted to residential buildings (Figure **11**, left). The measure becomes less efficient for the most extreme scenarios in 2050 and 2100, when the increase in TWL causes the surmounting of the barrier (Figure **10**, right). This assessment does not account for the impacts over those beach resorts and bathing facilities which are located along the barrier or between the barrier and the sea, and thus are equally exposed in both the baseline and the defended scenario; they would likely represent an additional 7-25% of the baseline damage.

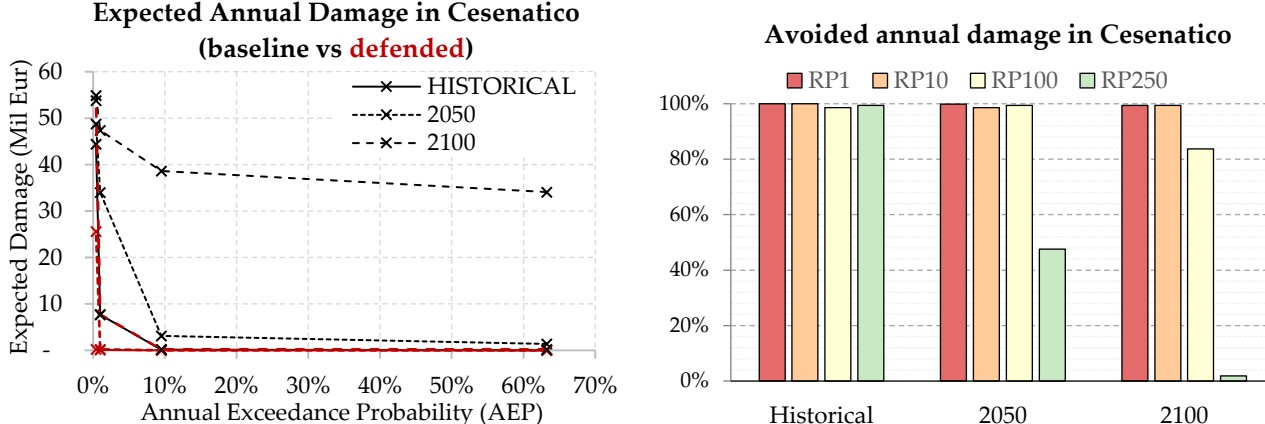

437

**Figure 10.** Cesenatico: Expected Annual Damage (EAD) according to undefended scenario up to 2100 [left];
EAD reduction thanks to hazard mitigation offered by the coastal barrier [right].

### 4.3 Cost-Benefit Analysis

The estimates of avoided direct flood impacts are accounted in a DRR-oriented CBA to evaluate the feasibility
of mitigation measures in terms of NPV, BCR and payback period for the two time-horizons (2021-2050: 30
years; and 2021-2100: 80 years). The assessment does not measure the indirect benefits brought in terms of
urban renovation, which are the primary focus of the *Parco del Mare* project, measuring, instead, only the direct
benefits in terms of direct flood damage reduction. In Figure **12**, the Expected Annual Benefits (EAB) brought
by defence measures grow at faster rate approaching 2100 in both sites, because of the larger expected damages
from increasing floods severity. The cost of defence implementation is repaid by avoided damage after about
40 years in Cesenatico and after 90 years in Rimini. At 2100, the BCR is 0.9 for Rimini and 1.8 for Cesenatico.
These results clearly indicate an overall profitability of the defence structure implementation over the long
term for Cesenatico. For the case of the municipality of Rimini, further investigation is required in order to
account for the non-DRR benefits of the seafront renovation project. For instance, the potential reduction in
indirect losses in terms of capital and labour productivity due to less frequent and less intense flooding events,
and the potential increase in tourism and well-being of citizens due to renewed urban landscape, are factors
that could be accounted for in a holistic CBA analysis and would likely return a shorter payback period.

**Table 2.** Summary of CBA for planned or designed seaside defence project in Rimini (all town / south section
only) and Cesenatico (all town / center only) over a time horizon of 30 and 80 years (2021 to 2050 and 2021 to
2100).

| | Rimini | | | | Cesenatico | | | |
|---|---|---|---|---|---|---|---|---|
| | *All town* | | *South only* | | *All town* | | *Center only* | |
| **Metrics** | 2050 | 2100 | 2050 | 2100 | 2050 | 2100 | 2050 | 2100 |
| Baseline EAD [M EUR] | 2.8 | 32 | 0.5 | 14.6 | 1.7 | 25.9 | 0.5 | 12.4 |
| Defended EAD [M EUR] | 2.4 | 17 | 0.1 | 0.9 | 0.1 | 0.4 | 0.1 | 0.4 |
| Expected Annual Benefits [M EUR] | 0.3 | 15 | 0.4 | 13.7 | 1.6 | 25.5 | 0.4 | 11.9 |
| Sum of EAB (discounted) [M EUR] | 5.6 | 30 | 4.1 | 27.8 | 12.0 | 79.4 | 4.7 | 28.6 |
| Sum of EAC (discounted) [M EUR] | 33.8 | 34.0 | 33.8 | 34.0 | 43.8 | 44.3 | 15.8 | 16.0 |
| Net Present Value [M EUR] | -28.3 | -4.0 | -29.8 | -6.3 | -31.8 | 35.1 | -11.24 | 12.6 |
| Benefit-Cost ratio [-] | 0.16 | 0.88 | 0.12 | 0.81 | 0.28 | 1.79 | 0.30 | 1.79 |

In order to better understand the potential benefits of the mitigation measures over different areas of the two
municipalities, we compare the results in terms of CBR over a selection of exposed records corresponding to
the town higher-density area (i.e. Cesenatico historical center). Table **2** summarizes the metrics of the
assessment for different area extent selections. CBA results do not differ much when considering different

extents. In Cesenatico benefits grow proportionally to costs, so that the payback time does not change when considering a section of the town or the whole coastal perimeter.

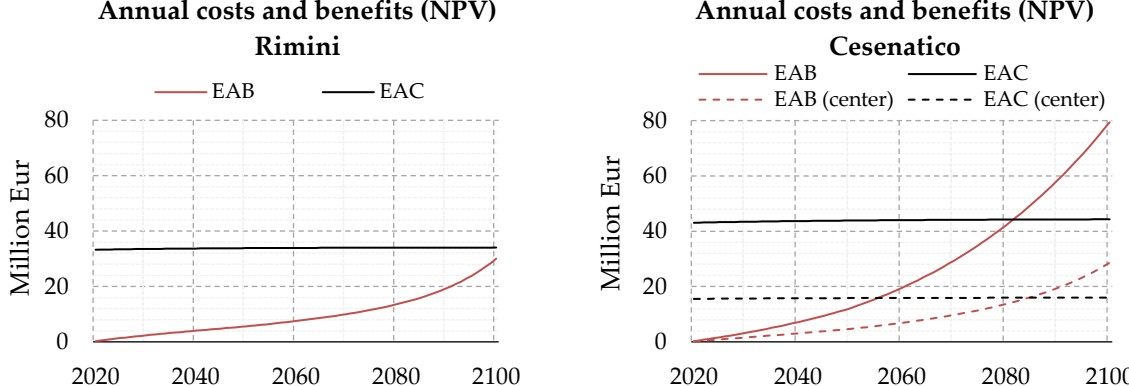

**Figure 11.** Cumulated flood defence costs and expected benefits at Net Present Value for Rimini (left) and Cesenatico (right).

## 5. Conclusion

In this study we addressed risk scenarios from coastal inundation over two coastal towns located along the North Adriatic coastal plain of Italy. This area is projected to become increasingly exposed to ESL events due to changes in MSL induced by SLR and local subsidence phenomena. Both locations are expected to suffer increasing economic losses from these events, unless effective coastal adaptation measures are put in place. In order to understand the upcoming impacts and the potential benefits of designed coastal projects, first we designed probabilistic ESL scenarios based on local historical observations; then, we projected these scenarios to 2050 and 2100, accounting for the combined effect of SLR and subsidence rates on the MSL. By using a high-resolution hydrodynamic model, we produced flood hazard maps associated with each ESL scenario under both the baseline and the "defended" hypothesis. The defended scenarios accounts for the effect of a coastal barriers based on the design of *Parco del Mare*, an urban renovation project under construction in Rimini. The same type of defence structure is envisaged along the coastal perimeter of the nearby town of Cesenatico. The hazard maps were fed to a locally-calibrated damage model in order to calculate the expected annual damage for both baseline and defended scenarios.

We run a CBA comparing expected damage in terms of flood losses over residential buildings, which represent the largest share of exposed buildings' footprints (93%). An increase in damage is expected for both urban areas from 2021 to 2100: in Cesenatico the EAD grows by a factor 96, in Rimini by a factor 49. The results show that profitability of present project investment grows over time in both locations, due to the increase of expected damage triggered by intense ESL events: the EAD under the baseline hypothesis is expected to increase by 3.5-fold in 2050, up to 10-fold in 2100. The benefits brought by the coastal defence project become much larger in the second half of the century: the EAB grows 6.1-fold in Rimini, 6.5-fold in Cesenatico, from 2050 to 2100. Avoided losses are expected to match the project implementation costs after about 40 years in Cesenatico and 90 years in Rimini. Benefits are found to increase proportionally to costs; the payback period in Cesenatico is the same considering either an investment on the protection of the whole town or only part of it.

Further assessments of these renovation projects should look to measure the indirect and spill-over effects over the local economy brought by the project, possibly accounting also for the intangible benefits and scenarios of exposure change. The results are calculated in relation to emission scenario RCP 4.5; compared to RCP 8.5 at 2050, the difference in SLR contribution is negligible (~0.05 m), while at 2100, the difference between

the two emission scenarios is larger (around 0.2 m), thus additional scenario analysis is suggested to better address risk by the end of the century. On the hazard modelling side, the particular consideration of combining wave setup and the swash component into a single wave action contribution component can be considered theoretically questionable, as wave setup is defined as the increase of mean sea level at the shore that is caused by the loss of wave momentum in the surf zone, being often referred as to the static component of wave runup. For future works facing a similar challenge, we recommended, given specific modelling limitations, to split wave action contribution as dynamic (i.e. swash) and static (runup) components.

## Data availability

Mattia Amadio, & Arthur H. Essenfelder (2021). Coastal flood inundation scenarios over Cesenatico and Rimini: hazard and risk for Business as Usual and Defended options [Data set]. Hosted by Zenodo: https://zenodo.org/record/4783443

## Authors contribution

MA, AHE and SB conceptualized the study and designed the experiments. AHE carried out the coastal hazard modelling. SR advised the model setup and calculation. SB and PM provided required data and expertise about the case study areas. MA performed the economic risk modelling and wrote the manuscript. SM supported the CBA calculations. JM and SB managed the funding acquisition and project supervision. All co-authors have reviewed the manuscript.

## Acknowledgment

The research leading to this paper received funding through the projects CLARA (EU's Horizon 2020 research and innovation programme under grant agreement 730482), SAFERPLACES (Climate-KIC innovation partnership) and EUCP – European Climate Prediction system under grant agreement 776613. We want to thank Luisa Perini for her kind support.

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

**Annex 1**
Here we present the equations and the graphical results of the theoretical ESL scenarios. TWL results from the
combination of storm surge, tide, and wave action contribution components, each following a general
functional form (i.e. harmonic component) describing the oscillation of water level, following trigonometric
functional forms for each component. The equations are given as follows.

$$T_l = T_{max} \times \cos\left(2\pi \frac{1}{T_p}(t + T_d + T_p)\right)$$   Eq. A1.1

Where *Tl* is the tide level in meters at any given time, *Tmax* is the maximum tide level in meters, *Tp* is the tidal
period in seconds, *Td* is the tidal period shift in time in seconds (used to match the peaks of tides and storm
surge events), and t is the time in seconds.

$$SS = SS_{max} \times 0.5 \times \left(1 + \cos\left(2\pi \frac{1}{S_p}(t + S_d + S_p)\right)\right)$$   Eq. A1.2

Where *SS* is the storm surge level in meters at any given time, *SSmax* is the maximum storm surge level in
meters, *Sp* is the storm surge duration in seconds, *Sd* is the storm surge shift in time in seconds (used to match
the peaks of tides and storm surge events), and t is the time in seconds.

$$W_{c,int} = 0.5 \times \left(1 + \cos\left(2\pi \frac{1}{S_p}(t + S_d + S_p)\right)\right)$$   Eq. A1.3

$$W_c = W_{max} \times 0.5 \times \left(1 + \cos\left(2\pi \frac{1}{W_p}\left(t + \frac{W_p}{4}\right)\right)\right) \times W_{c,int}$$   Eq. A1.4

Where *Wc* is the wave action contribution level in meters at any given time, *Wmax* is the maximum wave setup
level in meters, *Wp* is the wave period in seconds, *Wc,int* is the intensity factor [0-1] of the wave action
contribution event as a function of the storm surge intensity, *Sp* is the storm surge duration in seconds, *Sd* is
the storm surge shift in time in seconds (used to match the peaks of tides and storm surge events), and t is the
time in seconds.
We consider the wave action contribution component as a function of the intensity of the storm surge level, as
shown in Eqs. A1.3 and A1.4. As such, wave action contribution is simulated as composite function, where the
maximum wave action contribution level is designed to coincide in time with the maximum storm tide level,
and the directions of the waves are set to coincide with the direction of the storm surge event, in our case,
perpendicular to the coastline. This is done first to follow the assumption of worst-case scenario, and second
to incorporate the flood dynamics resulting from the momentum of waves directed inlands. The composite
function that combines Eqs. A1.1 to A1.4 and the effects of VLM and MSL (e.g. due to SLR) is shown in Eq.
A1.5. The results of each component in Eqs. A1.1 to A1.4 and for each probabilistic scenario are shown in
Figure A1.

$$TWL = MSL + VLM + T_l + SS + W_s$$   Eq. A1.5


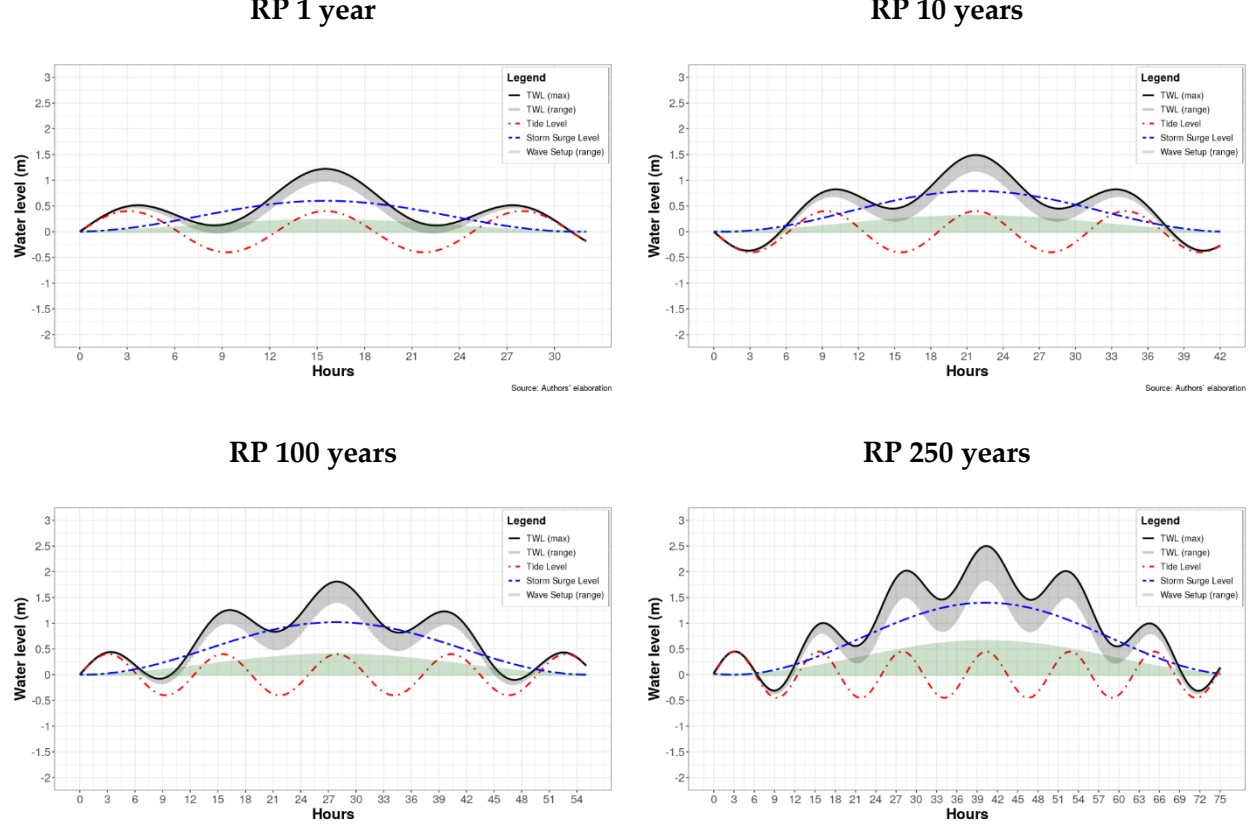

**Figure A1.** Dynamic boundary conditions for simulating theoretical extreme sea level events in ANUGA. The Total Water Level is show as the grey shaded area, while the maximum Total Water Level is shown by the black line, at any given time. The tide (dashed red line), storm surge (dashed blue line) and wave action contribution (green shaded area) components define the total water level. Configurations are shown for the return periods of once-in-1, 10, 100 and 250 years.

In order to verify the applicability of the aforementioned functions, we test the methods explained in this annex for all five ESL events that were observed along the coastline of the ER region during the year 2010, as reported in Perini et al. (2011). Observed sea level data is obtained from ISPRA, for the station Ravenna – Porto Corsini (Rete Mareografica Nazionale, 2021). We evaluate the goodness-of-fit of the methods by means of Coefficient of determination (R2). The results of this analysis are shown in Figure A2 below.

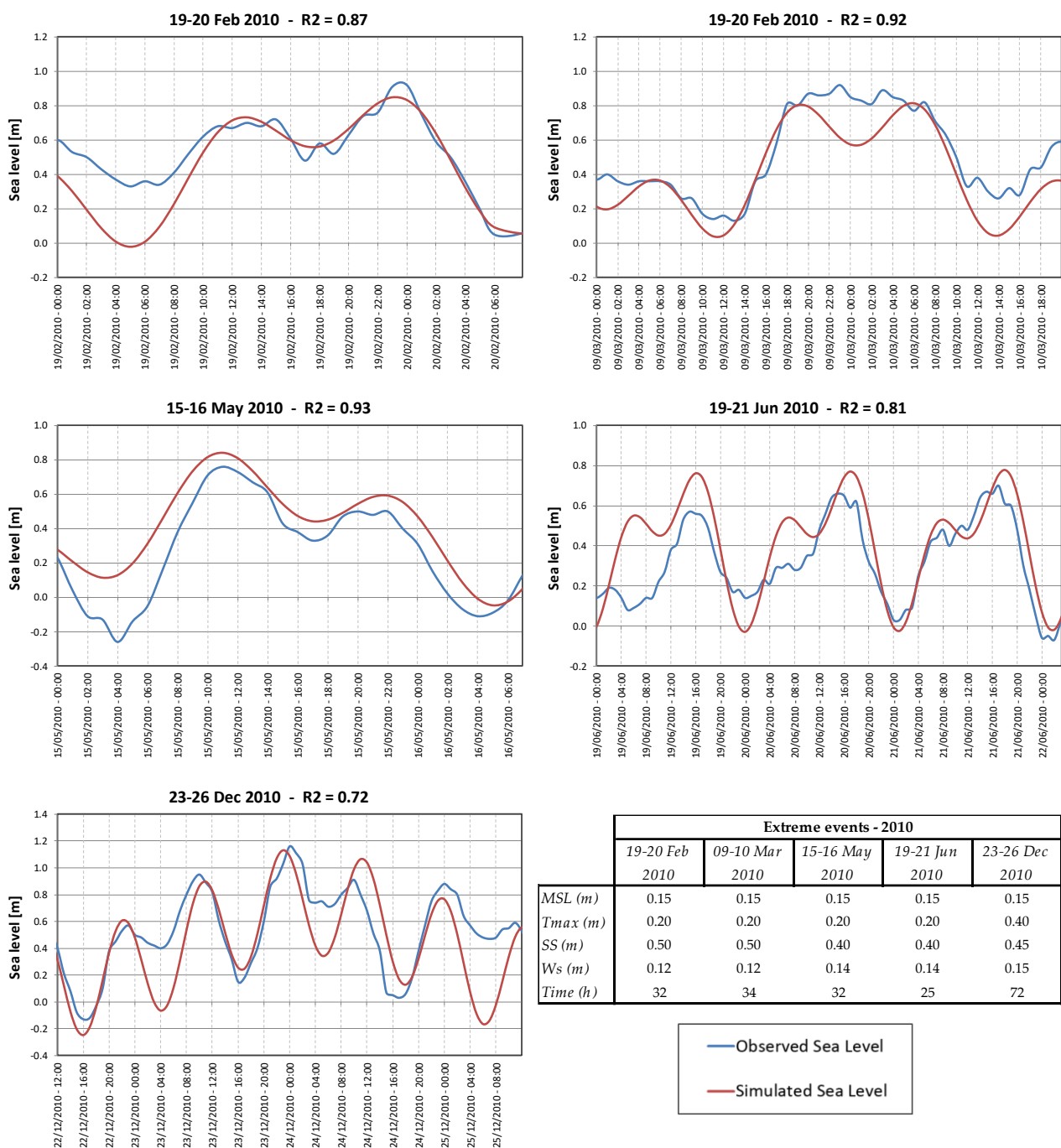

**Figure A2.** Comparison of the observed sea level (in blue) versus the simulated sea level using the harmonic components (in red).

**Annex 2**
A sensitivity analysis is carried out on the discount rate. Figure A2 below shows how the NPV changes with
discount rate *r* ranging from 1.5% to 5.5% (2020 to 2050) and 1% to 5% (2050-2100).

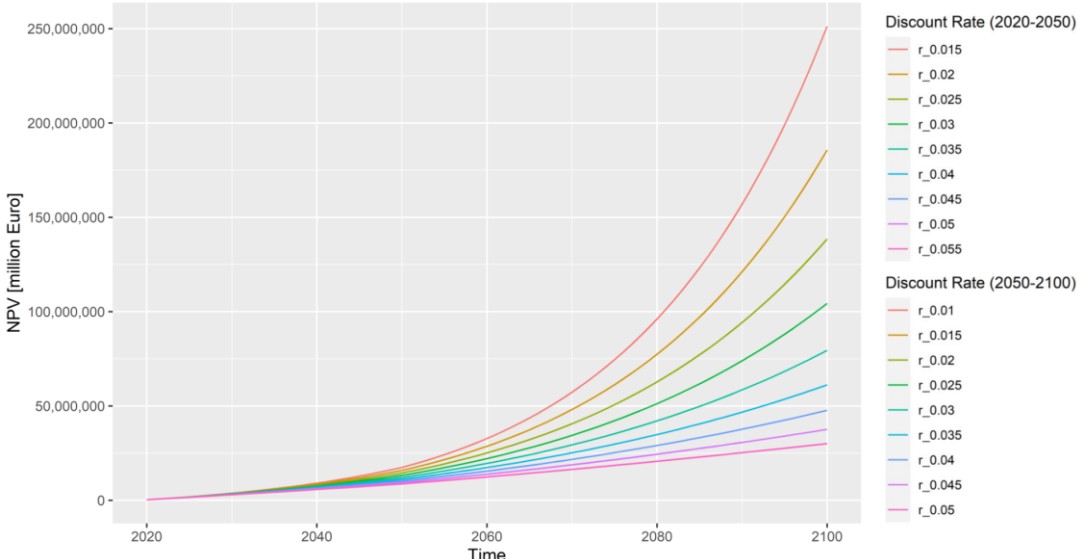

**Figure A2.** Sensitivity analysis of NPV using a variable discount rate.