# Peer review of "Cost-benefit analysis of coastal flood defence measures in the North"

_Natural Hazards and Earth System Sciences, 2020_

## Author Comment (AC1)

**Answer to reviewers**

We want to thank the reviewer for giving us the opportunity to submit an improved draft of the manuscript titled *Cost-benefit analysis of coastal flood defence measures in the North Adriatic Sea*. We appreciate the time and effort that you have dedicated to providing your valuable feedback on the manuscript. We have been able to incorporate changes to reflect most of the suggestions provided. We have enabled the track changes within the manuscript. The spelling and wording have been extensively revised following criteria of clarity, coherence and brevity.

Following is a point-by-point response to the reviewers' comments and suggestions.

**Reviewer #1**

**General comment:** I would clearly specify that Authors are referring to residential (or mixed) buildings only. It is only sometimes mentioned, while the scope of the manuscript appears to be a general cost-benefit analysis that includes other sectors too (infrastructures, industrial and commercial buildings, population, heritage and natural sites) and maybe also indirect damages. I would at least specify it in the introduction and remember it in the conclusions. Consider also to mention it also in the titles of paragraphs 3.7 and 4.2 and in the graphs' titles.

> ➢ Thank you for pointing this out, we clarified that only residential buildings are accounted in both the introduction and the conclusions, and specified that residential buildings represent 93% of total area.

**Introduction:** I suggest adding a paragraph about the situation in Italy, in order to introduce and justify the study area.

> ➢ We agree with your suggestion. We amended ch.2 adding more context and references regarding the situation of the Italian coast and the specific case study location.

**Ch. 2 and Sec. 3.1:** I would add some references, although the reader can find most of them in the flowing sections, but at a first read, they seem to be missing (e.g., about subsidence in the Padan plain at P2 L78-79 and VLM rates at P3 L102).

> ➢ Agreed, the same key references were added in both sections.

**P3 L85-86:** it is not clear to me how the sea level events' increasing is related to the socio-economic development of the coast. Please clarify.

> ➢ Thanks for pointing this out, the section has been amended and extended. The sentence was corrected:
>
> *The number of ESL events reported to cause impacts along the Emilia-Romagna coast shows a steady increase since the second half of the past century (Perini et al. 2011), which is in part explained by to the socio-economic development of the coast exposing increasing asset to flood risk.*

**P3 L93:** please provide some examples (years?) of coastal storms resulting in flooding of buildings and activities in the study area.

> ➢ Thank you for your valuable comment, we agree that more examples of flood impacts are needed to give better context, accordingly we extended ch.2 adding details about 3 recent dates, and some impact description from the latest (more severe) one.

**P8 L245-247:** I think the verb is missing, please correct.

> ➢ Thank you, we have fixed the sentence by adding the missing verb.

**P8 L249:** what about infrastructures and natural sites (mentioned on P3 L240)?

> ➢ We revised the explanation in ch 2: "The curve covers only residential and mixed-residential buildings, the area of which represents about 93% of total exposed footprints; other types (e.g. harbour infrastructures, industrial, commercial, historical monuments and natural sites) are excluded from risk computation."

**P8 L 259-261:** the statement about the validation of the depth-damage function on Italian empirical data is a repetition.

> ➢ Thanks, the sentence has been corrected.

**P9 L279:** please provide a reference or a justification to the 6 M€.

> ➢ Thank you for this important comment, the section about costs has been extended with all the justifications from official referenced sources in par. 3.8; numbers were also updated in the analysis.

**P9 L 292:** "costs" instead of "cots".

> ➢ Thanks, fixed.

**Figures 8-9 (left):** I would suggest plotting the defended scenarios too, so that the reader can immediately see the differences and be introduced to the right histograms.

> ➢ We agree on your suggestion, the two scenarios are plotted together in figure 8 and 9; however, it might be a bit difficult to read for Cesenatico, due to curves overlapping.

**P14 L388:** is the velocity as output needed? It is not used in the analysis…

> ➢ Thanks, you are correct, velocity is not used in our risk assessment framework. Accordingly, we removed it from the list of model outputs for the analysis.

**Appendix A** is never cited in the text. Please correct.

> ➢ Thank you, it is now mentioned in the section about discount rate.

---

## Author Comment (AC2)

**Answer to reviewer #2**

We want to thank the reviewer for giving us the opportunity to submit an improved draft of the manuscript titled *Cost-benefit analysis of coastal flood defence measures in the North Adriatic Sea*. We appreciate the time and effort that you have dedicated to providing your valuable feedback on the manuscript. We have been able to incorporate changes to reflect most of the suggestions provided. We have enabled the track changes within the manuscript. The spelling and wording have been extensively revised following criteria of clarity, coherence and brevity.

Following is a point-by-point response to the reviewers' comments and suggestions.

**Specific Comments**
**Line 52:** DRR project abbreviation is not defined. Please define for the readers' convenience.

> ➢ Thank you for pointing this out, full notation was added.

**Line 68:** Similarly, please define "ISTAT". If this is a reference, please include in the reference list.

> ➢ Agreed, ISTAT added as a reference.

**Line 78:** The authors suggest "… In addition to that, all the coastal profile of the Padan plain shows relatively fast subsiding rates, partially due to natural phenomena, but in large part linked to human activities." I believe this is land subsidence caused by landwater drainage. Please be more specific.

> ➢ That is true, and it is explained with more details in following par. 3.2:
>
> *Observed subsidence is about one order of magnitude faster where the aquifer system has been extensively exploited for agricultural, industrial and civil use since the post-war industrial boom. From the 1970s, however, with the halt of groundwater withdrawals, anthropogenic subsidence has been strongly reduced or stopped, but many of the induced effects still remain.*

**Line 85-87:** The authors state that ESL events are increasing due to socio-economic development of the coast? How? The socio-economic development could only affect the "impacts" of the ESLs, not ESLs as hazard levels. Please correct this sentence.

> ➢ Thank you for your most valuable comment, we agree that this was not clear, and the sentence has been amended.

**Line150:** The authors used RCP4.5 for the future projections of SLR. It would be interesting to see what happens with RCP8.5.

> ➢ Thank you for this suggestion. It would have been interesting to explore additional SLR scenarios, however the scenario analysis was constrained due to time and funding limitations, which in turn depend on the preferences of the stakeholders. As a result, it has been agreed with the Municipality of Rimini to simulate only one RCP scenario (RCP 4.5, selected as the "average"), but to consider a wider range of flood probabilities in term of return periods (i.e. 1, 10, 100, and 250). We think the one scenario that could be added in future work to extend recommendations is the RCP 8.5 at 2100, as the same scenario at 2050 does not differ significantly in terms of global SLR with respect to the RCP 4.5, while the scenario RCP 6.0 doesn't differ in terms of SLR significantly also at 2100.

[Figure]

**Line 150-152:** It is stated that "We consider the intermediate emission scenario RCP 4.5 (Thomson et al. 2011), projecting an increase in MSL of 0.53 m at 2100." Is this taken from the same reference, i.e., Thomson et al, 2011? I cannot find this in the relevant reference. Or is this calculated by the authors? If not please give the reference.

> Thomson reference only provides general description of the RCP4.5 emission scenario features, but all cited projections refer to previous reference (Vousdoukas et al 2017). We recognize this can be ambiguous, so Thomson reference was removed for clarity.

**Line 150-152:** Moreover, IPCC AR5 SLR projections give more local projected values of SLR. Apart from using a generic Global Mean Sea Level rise projection for the Mediterranean area, it would make more sense to account for more regional/local SLR values at specific coastal areas (as in IPCC AR5).

> Thank you for your valuable comment. We believe that in our analysis we do account for both local SLR historical observations at coast and specific downscaled projections for the central med basin. We have taken into consideration your comment and have amended par 3.3 to address these points more clearly:

*The long availability of tide gauge data along the N Adriatic coast allows to assess the changes in MSL in the last century. Records from the gauge station of Marina di Ravenna show an eustatic rise of 1.2 mm per year from 1890 to 2007, in good agreement with the eustatic rise measured at other stations in the Mediterranean Sea (Tsimplis and Rixen 2002; Carbognin et al. 2009). The projections of future MSL account for sea thermal expansions from four global circulation models, estimated contributions from ice-sheets and glaciers (Hinkel et al. 2014) and long-term subsidence projections (Peltier 2004). The ensemble mean is chosen to represent each RCP for different time slices. The increase in the central Mediterranean basin is projected to be approximately 0.2 m by 2050 and between 0.5 and 0.7 m by 2100, compared to historical mean (1970-2004) (Vousdoukas et al. 2017). We consider the intermediate emission scenario RCP 4.5, projecting an increase in MSL of 0.53 m at 2100. It must be noted that these projections, although downscaled for the Adriatic basin, do not account for the peculiar continental characteristics of the shallow northern Adriatic sector, where the hydrodynamics and oceanographic parameters partially depend on the freshwater inflow (Zanchettin et al. 2007).*

The dataset obtained from Vousdoukas et al (2017) - Extreme sea levels on the rise along Europe's coasts currently represents the best SLR estimate available for EU countries:

*Projections of RSLR indicate a statistically significant increase in MSL along the entire European coastline. The average RSLR across Europe is projected around 21 and 24 cm by the 2050s under RCP4.5 and RCP8.5, respectively. RSLR is projected to accelerate during the present century under both RCPs, reaching 53 and 77 cm by the year 2100 [...]. The RSLR projections show higher model agreement for the Mediterranean Sea and the Atlantic coast [...].*

[Figure]

**Figure 6.** *Time evolution of the 100-year $\eta_{w-ss}$ under Representative Concentration Pathway (RCP)4.5 and RCP8.5. Lines express the ensemble mean and colored patches the inter-model range (best-worst case). In order to understand better the spatial variations of the projections, the European coastline was divided in 10 geographical regions (see k), and values shown in (a–j) result from averaging all the information from each region.*

**Table 1.** Table Summarizing the Projected Absolute and Relative Changes of the 100-Year Event ESL (ΔESL and %ΔESL) Under RCP4.5 and RCP8.5, During the Years 2050 and 2100

|  | RCP45-2050 | | | RCP45-2100 | | | RCP85-2050 | | | RCP85-2100 | | |
|---|---|---|---|---|---|---|---|---|---|---|---|---|
| Area | ΔESL (m) | %Δ ESL | %Δ $\eta_{w-ss}$ | ΔESL (m) | %Δ ESL | %Δ$\eta_{w-ss}$ | ΔESL (m) | %ΔESL | %Δ $\eta_{w-ss}$ | ΔESL (m) | %Δ ESL | %Δ$\eta_{w-ss}$ |
| Black Sea | 0.25 | 18.6 | 7.9 | 0.60 | 44.2 | 4.2 | 0.27 | 19.7 | 5.0 | 0.81 | 60.1 | 1.1 |
| East Mediterranean | 0.20 | 14.3 | −5.9 | 0.53 | 38.8 | −1.1 | 0.22 | 16.0 | −6.7 | 0.71 | 52.3 | −8.6 |
| Central Mediterranean | 0.19 | 12.1 | −0.8 | 0.53 | 33.4 | 2.5 | 0.24 | 14.9 | 8.6 | 0.75 | 47.8 | −0.9 |
| West Mediterranean | 0.20 | 15.8 | −1.1 | 0.51 | 41.0 | −0.3 | 0.24 | 19.7 | 9.6 | 0.75 | 60.7 | 2.5 |
| S-North Atlantic | 0.18 | 4.9 | −13.8 | 0.48 | 12.9 | −17.8 | 0.18 | 4.9 | −29.2 | 0.66 | 17.8 | −19.4 |
| Bay of Biscay | 0.18 | 4.0 | −10.3 | 0.53 | 11.6 | −9.2 | 0.22 | 4.9 | −8.2 | 0.80 | 17.4 | −1.6 |
| N-North Atlantic | 0.27 | 4.7 | 28.9 | 0.64 | 11.3 | 13.8 | 0.29 | 5.2 | 27.5 | 0.88 | 15.7 | 17.0 |
| North Sea | 0.35 | 7.9 | 53.5 | 0.75 | 17.0 | 27.4 | 0.35 | 7.9 | 32.4 | 0.98 | 22.1 | 20.2 |
| Baltic Sea | 0.27 | 12.9 | 58.9 | 0.55 | 26.2 | 31.3 | 0.31 | 14.8 | 65.4 | 0.88 | 42.3 | 36.9 |
| Norwegian Sea | 0.21 | 5.1 | 18.1 | 0.46 | 11.2 | 9.5 | 0.23 | 5.7 | −0.8 | 0.64 | 15.4 | −5.8 |
| Europe | 0.25 | 8.3 | 18.5 | 0.57 | 19.4 | 9.5 | 0.27 | 9.2 | 15.7 | 0.81 | 27.3 | 7.1 |

ESLs, extreme sea levels; RCP, Representative Concentration Pathway.
%Δ$\eta_{w-ss}$ expresses how much of the projected change can be attributed to changes in extreme waves and storm surges, considering again the 100-year event.

(Vousdoukas et al. 2017)

**Line 177-179:** This is a minor comment, however, is there a need for using 1 m resolution for inland topography, while using 2m resolution for the coastal area, which is more significant for an inundation model?

> ➢ Unfortunately, this is the original availability of DEM resolution provided by the Ministry of Environment: the coastal DEM was produced at 2 m resolution, while the inland river network uses 1 m resolution.

**Line 228:** The authors mention "wave setup" and "wave runup" throughout the paper inconsistently. Although interconnected, these ESL contributors are different parameters. A clear definition is needed how wave setup OR wave runup was calculated for the coastal flooding purposes.

> ➢ Thanks for highlighting this, the wording has been fixed in text, tables and figures. We added explanation in par 3.6 that we only consider the Wave Setup component and included the limitation of 2D hydrodynamic models not being able to resolve vertical convection and breaking waves.

**Line 245:** The population is only accounted for the year 2011? Is this dataset also used for years 2050 and 2100? Please address this more clearly.

> ➢ We have amended text explaining that exposure is kept static in our assessment, thus allowing for a comparison focused on the effects of different hazard scenarios. Factors such as the potential increase in tourism and changes in urban mobility due to the urban renovation project could affect the population and urban growth dynamics at the future scenarios, but this is out of the scope of the current study.

**Line 280:** A reference is needed for "6M Eur per km" statement.

> ➢ Thank you for this important comment, the section about costs has been extended with all the justifications from official referenced sources in par. 3.8; numbers were also updated in the analysis.

**Line 285:** Please mention the sensitivity analysis for "r" values (which is Appendix A) conducted within this paragraph.

> ➢ Agreed, it is now mentioned in the section about discount rate (Annex 1).

**Line 310-311:** The authors state that "The north-western part and the marina are not affected by the coastal renovation project." Please specifically address this statement why and between which scenarios and years, referring to the figure 6.

> ➢ We made the sentence clearer:
>
> *The north-western part and the marina are outside of the defended area; these areas are therefore subject to a similar amount of flooding across scenarios (defended and undefended).*

**Line 363:** The authors found that the Benefit-Cost-Ratio is 0.82 for Rimini and claim that this is profitable. According to the definition they have given between lines 292-294, it is not, at least by the year of 2100, as it is lower than 1? Please reconsider this statement.

> ➢ We appreciate your valuable comment, which is indeed correct. We have amended the text by explaining that the Cost-Benefit-Analysis that has been performed is a DRR-specific analysis in the sense that it considers only the benefits of avoided direct impacts due to coastal floods. As included now in the manuscript, we argue that our results clearly indicate an overall profitability of the defence structure implementation over the long term for Cesenatico. For the case of the municipality of Rimini, further investigation is suggested in order to holistically account for the benefits of the seafront renovation project which would likely produce better CBA results:

*For instance, the potential reduction in indirect losses in terms of capital and labour productivity due to less frequent and less intense flooding events, and the potential increase in tourism and well-being of citizens due to renewed urban landscape, are factors that could be accounted for in a holistic CBA analysis and would likely return a shorter payback period.*

**Technical Comments:**
Line 94: "… occasions, …" Please add comma.
Line 129: "… industrial boom." Please delete the comma.
Line 189: "Figure 3. Prototype…" Please correct with the capital letter.
Line 293: "… and the costs; …" Please correct.
Line 359: "In figure 10 …" Please correct.

  ➢ All spelling and grammatical errors pointed out by the reviewers have been corrected.

We look forward to hearing from you regarding our submission and to respond to any further questions and comments you may have.

Sincerely,

Amadio Mattia

Essenfelder Arthur H.

---

## Referee Report (RR1)

Review for Cost-benefit analysis of coastal flood defence measures in the North Adriatic Sea by Mattia Amadio et al.

**General Comments:**

The paper was improved after the referee comments by appropriately addressing previous comments. However, there are still few issues to be addressed.

**Specific Comments:**

- In my opinion, it would be better if the reason choosing these two port sites is briefly mentioned in the last paragraph of introduction section.
- Line 138: Authors mention "…but many of the induced effects still remain" What are the remaining effects?
- Line 230-231: Authors state that "In our application, we estimate the TWL on the coastland at every timestep as the sum of extreme values for storm surge level (SS), wave setup (Ws), and max tide (Tmax), as shown in figure 4."
- Summation of extremes of surge and wave setup (RP10 etc) is not necessarily equal to the same return value for TWL, however, in Figure 4 and Table 1 it is implicated as such. Moreover, it is not clear how these RP values are calculated. Did authors use Extreme Value Analysis methods? If so, which ones? To determine the extreme sea levels (for TWLs), there are two common approaches:

i)       summing historical timeseries of tide+surge (and sometimes +wave setup) and then projecting the values with an Extreme Value Analysis method (see Coles, 2001 for EVA methods) [as in Vitousek et al 2017; Muis et al. (2016); Kirezci et al (2020); Rueda et al (2017)]

ii)      finding the pdf of each component of TWL and apply an ensemble Monte-Carlo approach (e.g. Vousdoukas et al 2018).

- Which dataset is used to calculate the surge, wave and tides?
- How did the authors determine the durations of extreme events in Table 1? Please clarify.
- How was Wave Setup calculated? What approach taken to determine wave setup? Additionally, which wave data is used, observed or hindcast? Please clarify.
- Figure 4 implicates a storm that the wave setup and storm surge both peak at the same time, which might not be the case in real life. How was this assumed? Please clarify.
- All in all, section 3.6, (the extreme sea level determination part) should be updated and the model inputs to determine the inundation should be explained in detail and clearly.
- In Table 1: Why is the RP250 tide value different the lower RP tides? Please clarify.
- Line 332: "surmounted" Do the authors mean overtopped or overflown?
- Line 346-348: "With less severe events (up to RP 100 years), the risk remains mostly confined around the marina area (outside the protection offered by the reinforced dune) producing an EAD below 10 thousand Eur; with more intense ESL scenarios (i.e. RP 250 years)" It is not clear to me what this statement means. Are not all the RPs included in EAD calculation (as in Figure 5)?

**Technical Comments:**

- Line 107: "Coastal inundation phenomena are caused…", please change to "Coastal inundation is caused…"
- Line 15: "N Adriatic", please change to North Adriatic.

---

## Author Response (AR2)

**Answer to reviewer**

We want to thank the reviewer for giving us the opportunity to submit an upgraded version of our manuscript titled *Cost-benefit analysis of coastal flood defence measures in the North Adriatic Sea*. We appreciate the comments on the method which allowed us to clarify some important points, in particular with regards to the presentation of the case study and the description of the data considered. We have been able to incorporate changes to reflect the valuable suggestions provided. The track changes are enabled within the manuscript to check all edits.

Following is a point-by-point response to the reviewers' comments and suggestions.

**Reviewer report #1**

**General Comments:** The paper was improved after the referee comments by appropriately addressing previous comments. However, there are still few issues to be addressed.

Thank you for your detailed feedback, we believe the manuscript improved by editing the text in agreement with your specific comments. The analytical method and the underlying data are now presented in a clearer, less ambiguous way.

**Specific Comments:**

- In my opinion, it would be better if the reason choosing these two port sites is briefly mentioned in the last paragraph of introduction section.
  - Paragraph from L55 has been extended mentioning the case study sites and the reason for the choice, that is the ongoing defence project in Rimini which could be later extended to Cesenatico:

We select two coastal cities as case study areas: i) Rimini, a touristic hotspot that is currently implementing a seafront renovation project; and ii) Cesenatico, a coastal city that could benefit from similar measures in addition to existing defence mechanisms.

- Line 138: Authors mention "...but many of the induced effects still remain" What are the remaining effects?
  - Sentence has been amended for clarity: induced effect refers to subsidence rates still ongoing around extraction areas that were active in the near past.
- Line 230-231: Authors state that "In our application, we estimate the TWL on the coastland at every timestep as the sum of extreme values for storm surge level (SS), wave setup (Ws), and max tide (Tmax), as shown in figure 4. Summation of extremes of surge and wave setup (RP10 etc) is not necessarily equal to the same return value for TWL, however, in Figure 4 and Table 1 it is implicated as such. Moreover, it is not clear how these RP values are calculated. Did authors use Extreme Value Analysis methods? If so, which ones? To determine the extreme sea levels (for TWLs), there are two common approaches:
  - summing historical timeseries of tide+surge (and sometimes +wave setup) and then projecting the values with an Extreme Value Analysis method (see Coles, 2001 for EVA methods) [as in Vitousek et al 2017; Muis et al. (2016); Kirezci et al (2020); Rueda et al (2017)]
  - ii. finding the pdf of each component of TWL and apply an ensemble Monte-Carlo approach (e.g. Vousdoukas et al 2018).

Thank you for your valuable comment. We did not perform Extreme Value Analysis of the data. Instead, we relied on previous regional studies that provide probabilistic analysis of extreme events conducted on the regional coast, for individual components (Perini et al. 2016, 2017). As explained next, this is the knowledge basis used to identify official flood hazard zones. From Perini:

Surge, tide and wave set-up were considered to define reference sea levels and, in the absence of statistical analysis for the combined return period, the worst-case scenarios for T1, T10 and T100 were assumed, considering the sum as the simultaneous occurrence of the three effects. The values used are from previous studies (tab. 1, Yu et al. 1994, Masina & Ciavola, 2011) and the comparison with real events recorded within the 'historical storm catalogue' has confirmed the validity of the method.

- Which dataset is used to calculate the surge, wave and tides?
  - > Section 3.6 has been amended and the choice of dataset is explained:

We obtain these variables from existing probabilistic analysis of extreme events conducted on the regional coast (Perini et al. 2016, 2017) and later adopted by the Regional Environmental Agency to define the official coastal flood hazard zones (ARPAE 2019).

We believe that using the same input used to define hazard perimeters (i.e. low, medium and high) is important to help the comparability of our study with official reports.

- How did the authors determine the durations of extreme events in Table 1? Please clarify.
  - > Text was amended by specify the reference study:

Additional details are wave period (Wp, in seconds) and event duration (Time, in hours), required to estimate the maximum extent of inland water propagation. Both variables are obtained from existing analysis of historical ESL events records, matched with the probabilistic distribution of RP scenarios (Armaroli et al. 2012; Armaroli and Duo 2018).

- How was Wave Setup calculated? What approach taken to determine wave setup? Additionally, which wave data is used, observed or hindcast? Please clarify.
  - As per the other components of TWL, wave setup was obtained from previous regional studies.
- Figure 4 implicates a storm that the wave setup and storm surge both peak at the same time, which might not be the case in real life. How was this assumed? Please clarify.
  - TWL represents the worst-case scenario, considering the max according to scenario probability for each of the components. As considered, our approach is precautionary as it provides worst-case scenario values.
- All in all, section 3.6, (the extreme sea level determination part) should be updated and the model inputs to determine the inundation should be explained in detail and clearly.

- Section 3.6 has been rewritten in agreement with your feedback. We thank you for providing these important comments and for the opportunity to improve the quality of our manuscript, and we hope that is now clearer.
- In Table 1: Why is the RP250 tide value different the lower RP tides? Please clarify.
  - The RP250 represents the most extreme scenario and as such also the Tide component is assumed to reach the max of the spring tidal range. This information has been clarified in the text.
- Line 332: "surmounted" Do the authors mean overtopped or overflown?
  - > Corrected with "overtopping".
- Line 346-348: "With less severe events (up to RP 100 years), the risk remains mostly confined around the marina area (outside the protection offered by the reinforced dune) producing an EAD below 10 thousand Eur; with more intense ESL scenarios (i.e. RP 250 years)" It is not clear to me what this statement means. Are not all the RPs included in EAD calculation (as in Figure 5)?
  - Thanks, the sentence was corrected and made clearer: Under less severe ESL scenarios (RP below 100 years), the risk remains mostly confined around the marina, which is located outside the defended area, producing an expected damage below 10 thousand Eur. Under more extreme ESL scenarios, the benefits of the Parco del Mare project protecting the southern part of Rimini become more evident, avoiding about 65% of the expected damages in the defended scenarios compared to the undefended ones.

**Technical Comments:**

- Line 107: "Coastal inundation phenomena are caused...", please change to "Coastal inundation is caused..."
- Line 15: "N Adriatic", please change to North Adriatic.
  - > Thank you, we fixed both sentences.

**Additional changes**

- Figure 8 and 9 (left) got fixed, with same x-axis scale.
- Additional language proof

---

## Author Response (AR3)

Dear Editor,

Thanks for your insightful review. Please find below our answer to each point raised.

1.  It is not clear how the information of figure 4 is used by the ANUGA model. Is it a boundary condition? The ANUGA model should compute the water level and not to adopt it from an external source. Please clarify.

➢   Thank you for your valuable comment. The information shown in Figure 4 aims to show how the boundary conditions in ANUGA are set-up during a theoretical storm tide event, in the case of Figure 4 for an example case of a once in 10 years return period. Each of the components shown in Figure 4 are considered independently, according to the intensity (i.e. water level) and duration of the respective storm tide event, and added to generate the black continuous line representing the storm tide level (or total water level, TWL) as a boundary condition in ANUGA. The detailed numeric information of the components considered is shown in Table 1. The flood model then takes into account the boundary conditions specified in Table 1 and dynamically shown in Figure 4 to simulate the dynamics within the domain, such as water flow through the canals and the flooding of land areas. For each return period scenario, a different idealised storm tide event is designed in order to match the maxima level of tide and surge (following the worst-case scenario assumption, as mentioned in line 60 and 239). We attach here these figures (also added as document annex1) representing each one of the dynamic scenarios considered for the different RPs.
Paragraph has been amended to clarify our approach.

[Figure]

2. The methodology of your simulations is not clear. It seems to me that you have simulated a set of idealized events, corresponding to different RPs under different VLM and local mean sea level hypothesis. Please clarify.

➢ Thank you for your comment. Yes, we simulated the coastal flood dynamics from a set of idealised storm tide events corresponding to the worst-case scenario, when extreme water level is the result of the combination of tides and storm surge. Such events do occur (e.g. Storm Xynthia in Western Europe), also along the coastline of the study area (e.g. extreme sea level event between the 15th and the 16th of November 2011 along the Emilia-Romagna coast). In this paper, we are not interested in forecasting water levels nor storm surge events, thus we do not provide information about the likelihood of such events happening along the coastline of the case study cities. Instead, we design scenarios that take into account the worst-case situation of coinciding storm tide events and evaluate the cost-benefits of flood defence measures against those worst-case scenario events. This is a traditional approach when designing engineering projects to safeguard public's safety, health and welfare.

3. The basis for the time evolution of the different components shown in figure 4 is not clear. They are strongly idealized (see previous comments). Please justify that they are representative of real events. Particularly the oscillations of the wave setup deserve an explanation. What is their cause? What is the corresponding evolution of the wave height?

➢ Thank you Thank you for your valuable comment. As discussed in the last comment, in the work developed in this paper, we are not interested in forecasting storm surge and spring tides along the coastline of the case study cities. As such, we assume a general functional form describing the oscillation of water level due to the different components defining extreme sea level, following trigonometric functional forms for each component. Each function, then, has a different period and magnitude, to reflect the contribution of each component. For instance, for simulating tides, we consider a function of period of 12 hours and 25 minutes and magnitude according to the scenario that we are considering. Trigonometric functions have been used with different degrees of complexity to approximate tidal levels and storm surge residuals (Boon 2004; Fuhrmann et al. 2019; Familkhalili et al. 2020).

Regarding the wave component, since ANUGA is a 2D model that cannot simulate wave breaking and the swash component of wave runup, we simulate just the wave setup component using data obtained from the literature. The wave setup is a relevant component of the total water level, and we incorporate it as a function of the intensity of the storm surge level, as shown in Figure 4. As such, wave setup is simulated as composite function, where the maximum wave setup level is designed to coincide with the maximum storm tide level, and the directions of the waves are set to coincide with the direction of the storm surge event, in our case, perpendicular to the coastline. This is done first to follow the assumption of worst-case scenario, and second to incorporate the flood dynamics resulting from the momentum of waves directed inlands.

A new paragraph has been included to 3.6, after table 1, to better explain our simulation approach. Figure 4 describing the simulation domain has been added.

4. For storm surges (height and duration), wave set up and tides, you use the results of Armaroli et al. 2012, Armaroli and Duo 2018. Please add a few sentences to describe the observational basis for these estimates (instrumentation used, its location and duration of the time series)

> Sentence amended: *Both variables are obtained from analysis of observations recorded during historical ESL events on the coast of Emilia-Romagna from 1946 to 2010 (Perini et al. 2011), matched with the probabilistic distribution of RP scenarios (Armaroli et al. 2012; Armaroli and Duo 2018).*

5. Table 1: The values of the column historical and TWL (2100) do not correspond to the sum of the components. Please use consistently the number of significant digits in the different columns. Abbreviations are explained in the text, but please add them to the caption to help the readers.

> Thanks, columns digits were made consistent.

**Other points to be changed, integrated or clarified**

i. Page 2 lines 65-66 and elsewhere in the text. Local RSL changes cannot be separated in a global eustatic component and the vertical land movement. Eustatic sea level changes are global sea level changes related to changes in the volume of water in the world ocean. The local sea level is affected by regional density and circulation changes, and mass addition (which is not spatially uniform). Please provide a clear conceptual framework for the components that you use for computing the local RSL rise.

> Sentence corrected. The manuscript has been revised to correct the terminology.

ii. Section 3.3 Observed and future sea level rise in the Adriatic is discussed in Lionello and Scarascia which estimated that the average sea level rise of the North Adriatic in the 20th century has been 1.3mm/year https://doi.org/10.1016/j.gloplacha.2013.03.004

> Sentence and references amended.

iii. Your answer to the reviewer comment asking for clarifying the reason for the difference between Tmax RP250 and shorter RPs is not clear to me. Is this difference based on previous studies or is it a subjective estimate? please clarify and justify the specific RP250 value.

> The value of Tmax in relation to RP250 is inferred from the data adopted by ARPA to define risk zones: in particular, in the referenced study from Perini et al 2016 (www.nat-hazards-earth-syst-sci.net/16/181/2016/) the spring tide is set as 0.45 m for worst case scenario (Table 1) and is unspecified for RP>100 years in Table 2, though it can only be 0.45 in order to consistently sum up to 2.50 m. Section 3.6 has been amended to clarify:

> *The high tide contribution grows from 0.40 m to 0.45 m, while wave setup near the shore ranges from 0.22 m to 0.65 m. We select the scenario RP 250 years as the upper boundary of hazard intensity, considering all components of TWL to reach their most extreme values and summing up to +2.5 meters over MSL.*

iv. Line 186 "change in trend" would mean acceleration or deceleration of an existing trend. Further it is not clear to which variable you are referring. I think you mean "... no strong evidence supporting a significant positive trend of marine storminess for the next future". In such a case a suitable citation would be https://doi.org/10.1016/j.gloplacha.2016.06.012

> Sentence amended, reference added.

v. Line 217 Section 3.6 please clarify briefly the difference between hydraulic, hydrodynamic and hydrostatic models or erase the sentence

> ➢ Sentence is removed.

vi.      Figure 5 : the meaning of the label "Factor" of the y-axis is not explained in the text. Further, it is not clear the relation between this figure and eq.(1)

> ➢ Figure is amended; caption is extended: *Schematic representation of the numerical integration of the damage function D(p) with respect to the exponential probability of the hazard events. Damage (Y axis) represents the ratio of damage to the total exposed value estimated up to the most extreme scenario (RP 250 years). Events with a probability of occurrence higher than once in a year are expected to not cause damage (grey area).*

**Other changes**

Corresponding author was added.

Annex 1 was added with figures of TWL boundary conditions for the model.

---

## Author Response (AR4)

Dear Editor,

We thank you for your relevant comments, which have led us to extensively revise the text in several sections and some figures. The updated version of the manuscript now stresses the need of dynamic flood modelling when performing probabilistic flood risk assessments, and it explains better how the components of TWL are considered in our modelling framework in relation to existing approaches. Our flood hazard analysis is based on a dynamic unfolding of coastal flood events using a simplified way to describe the TWL near shore, based on harmonic constituents. Harmonics constituents are the elements in a mathematical expression of a series of periodic terms and have been used in harmonic analysis for sea level prediction since the early 1960s. Even being a simplified method, our methodology allows for simulating the dynamics from nearshore to inland inundation, and we believe this to be a significant contribution to probabilistic flood risk assessments in coastal areas, as traditional approaches often make use of static (e.g. bathtub) approaches that do not take into account the dynamics of a coastal flood event and simply add the main components. Thus, we believe that our flood hazard methodology represents a better alternative to static methods.

In summary, the main changes are:

➢ We have extended the discussion on the state of the art regarding the choice of model and the components of the analysis in section 3.1. We now stress the fact that ESL scenarios under current conditions are designed on base of assessments performed in literature.

➢ We agree that Figure 4 (previously Figure 5) was unclear and open to misinterpretation, particularly in the representation of wave set-up contributing to TWL. In the previous version, in order to simplify the visualisation, wave contribution was (mis)represented as a continuous line with a frequency that did not present how this variable was actually accounted for in our modelling framework; instead, we have now represented wave contributions as a shaded area, stressing the relatively high-frequency contribution of such component to TWL.

➢ We included a new Annex 1 where we present the equations describing the harmonic components, plus a validation of the methods to observed events, and we have referred to the most recent studies that have adopted a similar approach.

➢ We have revised section 3.6 and divided the content into scenario design (now section 3.6) and inundation modelling (now section 3.7) to make easier the reading of the paper. Grey literature was removed.

➢ We thank you again for your valuable comments and those of the reviewers, as we believe that the quality of our manuscript has substantially increased after those exchanges. As such, we hope that our contribution to the scientific literature on probabilistic coastal flood risk modelling can be better appreciated now.

Amadio M.
Essenfelder A. H.

---

## Author Response (AR5)

Dear Editor,

Thanks again for your comments, which led to improve and clarify some of the methodological choices in our manuscript.

Regarding your last concern:

> *Specifically, equation Eq. A1.4 attributes to the wave set-up the same periodicity of the incoming breaking wave. This is really puzzling to me. The wave set-up is defined as the increase of mean sea level at the shore that is caused by the loss of wave momentum in the surf zone. The presence of such periodicity in your expression is not justified on the basis of the common definition of wave set-up. This point should be clarified or supported by a clear bibliographic reference.*

The literature from which we obtained (among others) the wave data (Perini et al 2016, based on Armaroli et al, 2009; Armaroli et al. 2012) accounts for wave runup+setup using a formula that considers the wave period (T):

1, 1-in-10, 1-in-100 years return period values of Table 2. Wave parameters were used to compute the run-up along each beach profile using the Holman (1986) formula modified by Komar (1998) to include the set-up (Eq. (2)):

$$R_{2\%}^T = 0.36g^{1/2}SH_\infty^{1/2}T \tag{2}$$

where $R_{2\%}$ is the 2% exceedance run up level, g is the acceleration of gravity, S is the beach slope, $H_\infty$ is the deep water wave height and T is the deep water wave period.

In our methodology, given the limitation of the 2D hydrodynamic model not resolving vertical convection and waves breaking (thus not allowing us to directly model wave run-up), we decided to include the wave motion intrinsic wave run-up as part of the wave set-up component, thus accounting for a periodicity equal to the incoming breaking waves. Indeed, calling the resulting variable wave set-up is theoretically questionable as wave set-up is considered as the static component of wave run-up (swash being considered the dynamic component). Even so, we believe that such can represent the motion of wave in a 2D hydrodynamic modelling framework, leading to a better consideration of the waves action on the resulting flood when compared to a simple static accounting of wave set-up.

Specifically, we propose the following edits to respond to your comment:

➢ add definitions of wave set-up and wave run-up and highlight the way in which wave action has been considered in literature and our manuscript; define the two combined variable as "*wave action contribution*", thus hopefully removing the ambiguity present in the previous version of the manuscript.

➢ reference the study where the wave runup/setup is computed considering period T (Armaroli et al. 2009; 2012).

➢ highlight the limitation of our wave-action approach in the conclusions, suggesting to explore in future works the chance to combine wave set-up and the swash component in a 2D-hydrodynamic model.

We thank you for your exhaustive revision of our manuscript (soon celebrating one year since the first submission), and we hope that such clarification is considered sufficient for the final acceptance of our manuscript.

Best regards

Amadio M., Essenfelder A. H.

---

## Author Response (AR6)

Dear Editor,

thanks for the final acceptance of our manuscript,

we corrected the "wave action" term as suggested using instead "action of waves" and "wave contribution". Charts legends were updated accordingly.

Annexes 1 and 2 has been renamed "Appendix A" and "Appendix B" and figure captions were renamed accordingly.

Best regards

Amadio M., Essenfelder A. H.